# DYNAMIC PROMPT LEARNING VIA POLICY GRADIENT FOR SEMI-STRUCTURED MATHEMATICAL REASONING

**Pan Lu**[1,3], **Liang Qiu**[1], **Kai-Wei Chang**[1], **Ying Nian Wu**[1], **Song-Chun Zhu**[1],
**Tanmay Rajpurohit**[2], **Peter Clark**[3], **Ashwin Kalyan**[3]
[1]University of California, Los Angeles, [2]Georgia Institute of Technology, [3]Allen Institute for AI

## ABSTRACT

Mathematical reasoning, a core ability of human intelligence, presents unique challenges for machines in abstract thinking and logical reasoning. Recent large pre-trained language models such as GPT-3 have achieved remarkable progress on mathematical reasoning tasks written in text form, such as math word problems (MWP). However, it is unknown if the models can handle more complex problems that involve math reasoning over heterogeneous information, such as tabular data. To fill the gap, we present Tabular Math Word Problems (TABMWP), a new dataset containing 38,431 open-domain grade-level problems that require mathematical reasoning on both textual and tabular data. Each question in TABMWP is aligned with a tabular context, which is presented as an image, semi-structured text, and a structured table. There are two types of questions: *free-text* and *multi-choice*, and each problem is annotated with gold solutions to reveal the multi-step reasoning process. We evaluate different pre-trained models on TABMWP, including the GPT-3 model in a few-shot setting. As earlier studies suggest, since few-shot GPT-3 relies on the selection of in-context examples, its performance is unstable and can degrade to near chance. The unstable issue is more severe when handling complex problems like TABMWP. To mitigate this, we further propose a novel approach, PROMPTPG, which utilizes policy gradient to learn to select in-context examples from a small amount of training data and then constructs the corresponding prompt for the test example. Experimental results show that our method outperforms the best baseline by 5.31% on the accuracy metric and reduces the prediction variance significantly compared to random selection, which verifies its effectiveness in the selection of in-context examples.[1]

## 1 INTRODUCTION

Developing machines equipped with mathematical reasoning capabilities is one of the long-standing goals of artificial intelligence. Solving math word problems (MWPs) is a well-defined task to diagnose the ability of intelligent systems to perform numerical reasoning and problem-solving as humans. A surge of datasets has been proposed to facilitate the research in this domain (Upadhyay & Chang, 2017; Amini et al., 2019; Miao et al., 2020; Cobbe et al., 2021). However, most existing MWP datasets focus on textual math word problems only. Tables, widely distributed in different documents such as invoices, health records, and financial reports, contain rich structured information different from unstructured text. Solving math word problems in such a tabular context is much more challenging than existing MWP benchmarks since the system needs to make cell selections and align heterogeneous information before performing further numerical reasoning.

To fill this gap, we propose Tabular Math Word Problems (TABMWP), a new large-scale dataset that contains 38,431 math word problems with tabular context, taken from grade-level math curricula. There are two question types: *free-text* questions in which the answer is an integer or decimal number, and *multi-choice* questions where the answer is a text span chosen from option candidates. Different from existing MWP datasets, each problem in TABMWP is accompanied by a tabular context, which is represented in three formats: an image, a semi-structured text, and a structured

---

[1]The data and code are available at `https://promptpg.github.io`.
Work was partially done while Pan Lu was an intern at Allen Institute for AI (AI2).

| | |
|---|---|
| square beads | $2.97 per kilogram |
| oval beads | $3.41 per kilogram |
| flower-shaped beads | $2.18 per kilogram |
| star-shaped beads | $1.95 per kilogram |
| heart-shaped beads | $1.52 per kilogram |
| spherical beads | $3.42 per kilogram |
| rectangular beads | $1.97 per kilogram |

**Question:** If Tracy buys 5 kilograms of spherical beads, 4 kilograms of star-shaped beads, and 3 kilograms of flower-shaped beads, how much will she spend? (unit: $)
**Answer: 31.44**
**Solution:**
Find the cost of the spherical beads. Multiply: $3.42 × 5 = $17.10.
Find the cost of the star-shaped beads. Multiply: $1.95 × 4 = $7.80.
Find the cost of the flower-shaped beads. Multiply: $2.18 × 3 = $6.54.
Now find the total cost by adding: $17.10 + $7.80 + $6.54 = **$31.44**.
She will spend $**31.44**.

| Sandwich sales | | |
|---|---|---|
| **Shop** | **Tuna** | **Egg salad** |
| City Cafe | 6 | 5 |
| Sandwich City | 3 | 12 |
| Express Sandwiches | 7 | 17 |
| Sam's Sandwich Shop | 1 | 6 |
| Kelly's Subs | 3 | 4 |

**Question:** As part of a project for health class, Cara surveyed local delis about the kinds of sandwiches sold. Which shop sold fewer sandwiches, Sandwich City or Express Sandwiches?
**Options:** (A) Sandwich City (B) Express Sandwiches
**Answer: (A) Sandwich City**
**Solution:**
Add the numbers in the Sandwich City row. Then, add the numbers in the Express Sandwiches row.
Sandwich City: 3 + 12 = 15. Express Sandwiches: 7 + 17 = 24.
15 is less than 24. **Sandwich City** sold fewer sandwiches.

Figure 1: Two examples from the TABMWP dataset. The example above is a *free-text* problem with a numerical answer; the example below is a *multi-choice* problem with a textual answer.

table. Each problem is also annotated with a detailed solution that reveals the multi-step reasoning steps to ensure full explainability. To solve problems in TABMWP, a system requires multi-hop mathematical reasoning over heterogeneous information by looking up table cells given textual clues and conducting multi-step operations to predict the final answer. Take the problem above in Figure 1 as an example. To answer the question "*how much will she spend (if Tracy buys three kinds of beads)*?", we first need to look up the corresponding three rows in the given table, calculate the individual cost for each kind of bead, and finally sum three costs up to get the answer of 31.44.

Inspired the success of the large pre-trained language model GPT-3 (Brown et al., 2020) in solving math word problems (Wei et al., 2022; Wang et al., 2022), we first build a strong baseline using few-shot GPT-3 on TABMWP. A few in-context examples are randomly selected from the training set, along with the test example, and are constructed as a prompt for GPT-3 to predict the answer. However, recent studies have shown that this type of few-shot learning can be highly unstable across different selections of in-context examples (Zhao et al., 2021; Liu et al., 2022a; Lu et al., 2022c). It could be worse on TABMWP since its problems are distributed across multiple question types and diverse table layouts. Liu et al. (2022a) try to address this issue by retrieving semantically similar examples. However, this method might not work well sometimes on TABMWP because it is not capable of measuring the similarity of structured information, such as the number of cells in tables.

To alleviate this challenge, we further propose a novel approach that can learn to select in-context examples from a small amount of training data via policy gradient for prompt learning, termed PROMPTPG. As illustrated in Figure 2, an agent learns to find optimal in-context examples from a candidate pool, with the goal of maximizing the prediction rewards on given training examples when interacting with the GPT-3 environment. A policy network defines the strategy of how to select the in-context examples given the current training example. The policy network is built on top of the language model BERT (Devlin et al., 2018) with fixed parameters, followed by a one-layer linear neural network with learnable parameters. The learnable parameters are updated following the policy gradient strategy (Sutton et al., 1998). Unlike random selection (Wei et al., 2022; Wang et al., 2022), brute-force search, or retrieval-based selection (Liu et al., 2022a), PROMPTPG learns to construct the prompt dynamically given the candidate pool when interacting with the GPT-3 API.

We implement two state-of-the-art methods as baselines, i.e., UnifiedQA (Khashabi et al., 2020) on general question answering and TAPEX (Liu et al., 2022b) on tabular question answering. Both are implemented in pre-trained and fine-tuned settings. Experimental results show that our model PROMPTPG can achieve an overall accuracy of 68.23% on TABMWP, which greatly surpasses previous methods by a large margin of up to 5.31%. Further analysis demonstrates that PROMPTPG can select better in-context examples compared with a wide range of existing selection strategies and reduce the prediction variance significantly compared to random selection.

The main contributions of our work are as follows: (a) We present a new large-scale dataset, TABMWP, the first dataset for math word problems with tabular context; (b) We propose a novel

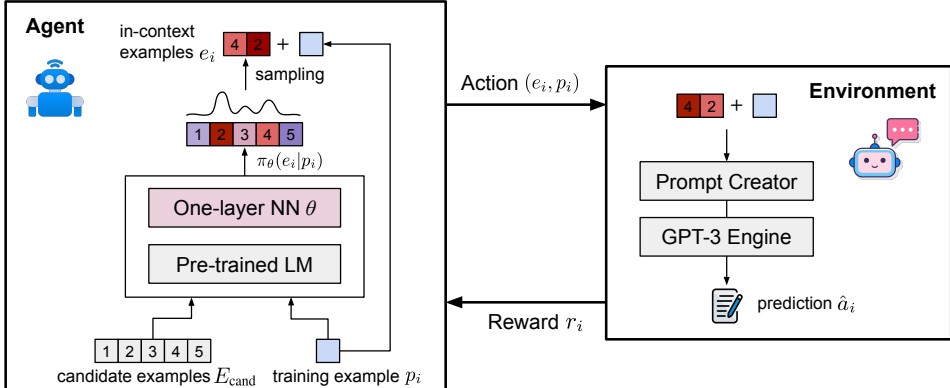

Figure 2: Our proposed PROMPTPG is able to learn to select performing in-context examples via policy gradient when interacting with the GPT-3 API without any manually designed heuristics.

approach, PROMPTPG, which learns the prompt dynamically via policy gradient to select in-context examples for few-shot GPT-3. To the best of our knowledge, it is the first work that applies reinforcement learning to select in-context examples for the few-shot GPT-3 model; (c) Experimental results show that PROMPTPG achieves an improvement of up to 5.31% on TABMWP over existing methods, with reduced selection instability compared to random selection.

## 2 THE TABMWP DATASET

### 2.1 TASK FORMULATION

A tabular math word problem $p$ is represented as a pair $(t, q)$, where $t$ is a table context and $q$ is a question. The table $t$ could be represented in a visual format as an image, semi-structured text, or a structured database. In this work, we focus on the semi-structured format as the table context for simplicity. The table $t$ features complicated layouts and formats: it contains multiple rows and columns, and each cell can be a string of text, a string of a number, or a mix of them. Depending on the question and answer types, the question $q$ may be accompanied by multiple-choice options $c = \{c_1, c_2, \ldots, c_n\}$ or a unit $u$. Given a semi-structured tabular context $t$ and an unstructured question text $q$, the task is to generate the answer $a$, which is either numerical only text for a *free-text* question, or a text span from given options for a *multiple-choice* question.

### 2.2 DATASET CONSTRUCTION

**Data collection.** We construct TABMWP based on openly available content and more details are provided in Appendix A.1. Only math word problems that are accompanied by a tabular context and a detailed solution are collected. We develop a script to extract the tabular context, the question, options that apply, the correct answer, and the solution for each problem. These elements can be precisely identified using HTML tags. For each table, we take a screenshot and store its raw text.

**Data preprocessing.** To make TABMWP compatible with various baselines, we represent the tabular context as three formats: an image, *semi-structured* text, and a *structured* spreadsheet. The semi-structured format is created by converting the raw table text into a flattened token sequence, with each row separated by a newline character '\n' and each column separated by '|'. The semi-structured text is further transformed to the structured format, which can be easily retrieved and executed by SQL-based methods (Liu et al., 2022b) using packages like `pandas`. For clarity, the table title is separated from the raw table. Examples of three formats are shown in Appendix A.1.

For better quantitative evaluation, we formalize the TABMWP problems as two question types: (a) *free-text* questions, where the answer is numerical text only and the unit text is separately extracted; and (b) *multi-choice* questions, the answer of which is the text span from choice options. The order of choice options is shuffled to alleviate distribution bias. Redundant information in solutions is removed, and some solutions are manually rewritten to be more human-readable. Finally, problems with the same table, question, and answer text are regarded as redundant and thus removed. We further conduct quality control to ensure data quality, which is discussed in Appendix A.1.

## 2.3 DATASET STATISTICS

**Key statistics.** The TABMWP dataset contains 38,431 tabular math word problems, which are partitioned with 6:2:2 into the training, development, and test splits, corresponding to 23,059, 7,686, and 7,686 problems. Their main statistics are shown in Table 1. 74.7% of the questions in TABMWP belong to *free-text* questions, while 25.3% are *multi-choice* questions. There are 28,876 different questions, 6,153 different answers, and 35,442 different solutions, indicating that TABMWP has a rich diversity in the problem distribution. The questions have an average of 22.1 words in length and solutions of 49.5, showing that they have lexical richness.

One distinct characteristic of TABMWP is that each problem is accompanied by a tabular context, without which the problem would be unsolvable. There are 37,644 different tables in total, and 60.5% of the tables have a title. The table has an average of 5.9 rows and 2.2 columns, which results in an average of 12.9 cells and a maximum of 54 cells. These statistics suggest that tables in TABMWP distribute diversely across semantics and layouts.

| Statistic | Number |
|---|---|
| Total questions | 38,431 |
| * *free-text* questions | 28,719 |
| * *multi-choice* questions | 9,712 |
| # of different questions | 28,876 |
| # of different answers | 6,153 |
| # of different solutions | 35,442 |
| # of different tables | 37,644 |
| # of tables with a title | 23,259 |
| # of table cells (Average/Max) | 12.9 / 54 |
| # of table rows (Average/Max) | 5.9 / 11 |
| # of table columns (Average/Max) | 2.2 / 6 |
| Question length (Average/Max) | 22.1 / 92 |
| Answer length (Average/Max) | 1.1 / 27 |
| Solution length (Average/Max) | 49.5 / 350 |

Table 1: Key statistics for TABMWP.

**Comparison to existing datasets.** As shown in Table 2, TABMWP differs from related datasets in various aspects: (1) TABMWP is the first dataset to study math word problems over tabular context on open domains and is the largest in terms of data size; (2) Problems in TABMWP are annotated with the tabular context, unlike previous MWP datasets in the first segment; (3) Different from Table QA datasets like FinQA, TAT-QA, and MultiHiertt, a lack of either mathematical reasoning or the tabular context renders the problems in TABMWP unanswerable; (4) There are two question types in TABMWP, and the answer could be a text span, an integer number, or a decimal number; (5) Each problem is annotated with natural language solutions to reveal multi-hop reasoning steps.

| Dataset | Size | #Table | Need Math? | Need Table? | Table Type Domain | Table Type Format | Question Type Free-text | Question Type MC | Answer Type Text | Answer Type Integer | Answer Type Decimal | Solution Type |
|---|---|---|---|---|---|---|---|---|---|---|---|---|
| Dolphin18K (2016) | 831 | ✗ | ✓ | ✗ | ✗ | ✗ | ✓ | ✗ | ✗ | ✓ | ✓ | formula |
| DRAW-1K (2017) | 1,000 | ✗ | ✓ | ✗ | ✗ | ✗ | ✓ | ✗ | ✗ | ✓ | ✓ | formula |
| Math23K (2017) | 23,162 | ✗ | ✓ | ✗ | ✗ | ✗ | ✓ | ✗ | ✗ | ✓ | ✓ | formula |
| MathQA (2019) | 37,297 | ✗ | ✓ | ✗ | ✗ | ✗ | ✗ | ✓ | ✗ | ✓ | ✓ | formula |
| ASDiv (2020) | 2,305 | ✗ | ✓ | ✗ | ✗ | ✗ | ✓ | ✗ | ✓ | ✓ | ✓ | formula |
| SVAMP (2021) | 1,000 | ✗ | ✓ | ✗ | ✗ | ✗ | ✓ | ✗ | ✗ | ✓ | ✗ | formula |
| GSM8K (2021) | 8,792 | ✗ | ✓ | ✗ | ✗ | ✗ | ✓ | ✗ | ✗ | ✓ | ✗ | text |
| IconQA (2021b) | 107,439 | ✗ | ✓ | ✗ | ✗ | ✗ | ✓ | ✓ | ✓ | ✓ | ✗ | ✗ |
| FinQA (2021) | 8,281 | 2,766 | ✓ | 76.6% | finance | text | ✓ | ✗ | ✗ | ✓ | ✓ | program |
| TAT-QA (2021) | 16,552 | 2,747 | 50.0% | ✓ | finance | text | ✓ | ✗ | ✗ | ✓ | ✓ | ✗ |
| MultiHiertt (2022) | 10,440 | 9,843 | ✓ | 89.8% | finance | text | ✓ | ✗ | ✗ | ✓ | ✓ | ✗ |
| **TABMWP (ours)** | **38,431** | **37,644** | ✓ | ✓ | **open** | **text*** | ✓ | ✓ | ✓ | ✓ | ✓ | **text** |

Table 2: A comparison of MWP and Table QA datasets that require numerical reasoning. *text*: each table in TABMWP is accompanied by an image format.

## 3 METHODS

### 3.1 FEW-SHOT GPT-3 FOR TABMWP

Provided with a few in-context examples of math word problems as the context, GPT-3 can generate the answer for a test problem, and shows impressive performance across different MWP datasets (Wei et al., 2022; Wang et al., 2022). Inspired by its success, we first build a strong baseline using few-shot GPT-3 on our TABMWP dataset. Specifically, a few training examples, along with the test example $p_i$, are provided to GPT-3 for the answer prediction. Each training example consists of a table context $t$, a question $q$, options $c$ that apply, and an answer $a$. To make the few-shot

GPT-3 model workable on TABMWP, we utilize the semi-structured format as the tabular context. Following Wei et al. (2022), a solution $s$ can be augmented in front of the answer $a$ to reveal the multi-step reasoning process, which is able to boost the prediction performance.

## 3.2 DYNAMIC PROMPTING VIA POLICY GRADIENT

The in-context examples can be randomly (Wei et al., 2022; Wang et al., 2022) or retrieval-based selected (Liu et al., 2022a) from the training set. Recent research, however, has shown that few-shot GPT-3 can be highly unstable across different selections of in-context examples and permutations of those examples (Zhao et al., 2021; Liu et al., 2022a; Lu et al., 2022c). This instability may be more severe on TABMWP, where examples are more distinct because they include both unstructured questions of various types and semi-structured tables in various layouts. To alleviate this issue, we aim to propose a novel approach that can learn to select performing in-context examples using a policy gradient strategy, without brute-force searching or manually designed heuristics.

Formally, given a TABMWP problem $p_i$, we want the agent to find $K$ in-context examples $e_i = \{e_i^1, e_i^2, ..., e_i^K\}$ from a candidate pool $E_{\text{cand}}$, and generate the answer $\hat{a}_i$, maximizing a reward $r_i = R(\hat{a}_i|p_i)$. The in-context examples are selected according to a policy

$$e_i^k \sim \pi_\theta(e_i|p_i),\ e_i^k \in E_{\text{cand}}, e_i^k \text{ are independent for } k = \{1, 2, ..., K\}, \tag{1}$$

where $\theta$ are the policy's parameters. The answer is generated through: $\hat{a}_i = \text{GPT-3}(e_i, p_i)$ using the selected examples and the given problem as the input prompt. The reward is then computed by evaluating the generated answer $\hat{a}_i$ with respect to the ground truth answer $a_i$:

$$r_i = R(\hat{a}_i|p_i) = \text{EVAL}(\hat{a}_i, a_i),\ r_i \in \{-1, 1\}. \tag{2}$$

The function $\text{EVAL}()$ returns a reward of 1 if the generated answer aligned with the label and $-1$ otherwise. Our goal is to maximize the expected reward of the generated answer under the policy $\mathbb{E}_{e_i \sim \pi_\theta(e_i|p_i)}[R(\text{GPT-3}(e_i, p_i))]$. We optimize the reward with respect to the parameters of the policy network using the Policy Gradient method (Sutton et al., 1998). The expected reward cannot be computed in closed form, so we compute an unbiased estimation with Monte Carlo Sampling,

$$\mathbb{E}_{e_i \sim \pi_\theta(e_i|p_i)}[R(\text{GPT-3}(e_i, p_i))] \approx \frac{1}{N} \sum_{i=1}^{N} R(\text{GPT-3}(e_i, p_i)),\ e_i \sim \pi_\theta(e_i|p_i), \tag{3}$$

where $N$ is the size of each batch yielded from our training problem set $P_{\text{train}}$. In this work, we experiment using the REINFORCE policy gradient algorithm (Williams, 1992):

$$\nabla\mathbb{E}_{e_i \sim \pi_\theta(e_i|p_i)}[R(\text{GPT-3}(e_i, p_i))] = \mathbb{E}_{e_i \sim \pi_\theta(e_i|p_i)}\nabla_\theta \log(\pi_\theta(e_i|p_i))R(\text{GPT-3}(e_i, p_i))$$
$$\approx \frac{1}{N} \sum_{i=1}^{N} \nabla_\theta \log(\pi_\theta(e_i|p_i))R(\text{GPT-3}(e_i, p_i)),\ e_i \sim \pi_\theta(e_i|p_i). \tag{4}$$

Intuitively, if the predicted answer is correct, we update the policy so that the probability of selecting the same prompts gets higher. Otherwise, we update the policy to reduce the probability of selecting such less matched examples. The learning process is summarized in Algorithm 1 in the appendix.

To get the contextualized representation of the given problem and candidate examples, we use the BERT (Devlin et al., 2018) `[CLS]` token representation as the problem encoding. We add a small linear layer on top of the BERT final pooling layer. That allows our model to learn both the semantic similarity that the pre-trained BERT model provides and the hidden logical similarity shared among the math problems. During training, the parameters of BERT are fixed and only the appended linear layer is updated, i.e., $\theta$ is composed of the learnable parameters $\mathbf{W}$ and $\mathbf{b}$:

$$\mathbf{h}(e_i) = \mathbf{W}(\text{BERT}(e_i)) + \mathbf{b},$$
$$\mathbf{h}(p_i) = \mathbf{W}(\text{BERT}(p_i)) + \mathbf{b},$$
$$\pi_\theta(e_i|p_i) = \frac{\exp[\mathbf{h}(e_i) \cdot \mathbf{h}(p_i)]}{\sum_{e_i' \in E_{\text{cand}}} \exp[\mathbf{h}(e_i') \cdot \mathbf{h}(p_i)]}. \tag{5}$$

## 4 EXPERIMENTS

### 4.1 EXPERIMENTAL SETTINGS

**Baselines.** We first develop two large language models, UnifiedQA (Khashabi et al., 2020) and TAPEX (Liu et al., 2022b), in both pre-trained and fine-tuned settings, as strong baselines on TABMWP. Different model sizes are included to examine the performance across different model capacities. We further implement the zero-shot GPT-3 model, the few-shot GPT-3 model, and their chain-of-thought (CoT) reasoning variants (Wei et al., 2022). We also study the heuristic guess baseline and human performance to analyze the lower and upper bounds on TABMWP, respectively.

**Evaluation metric.** The answer part is extracted from the GPT-3 generation using manually designed regular expressions. To evaluate the baselines and our method, we utilize the accuracy metric to determine if the generated answer is correct given the ground truth answer. For *free-text* problems where the answer is set as a number, we normalize the prediction and the label to decimal numbers with two-digit precision and check if their values are equivalent. For *multi-choice* problems, we choose the most similar one from options to the generated answer following Khashabi et al. (2020).

**Implementation details.** Fine-tuned UnifiedQA and TAPEX baselines are trained on the train split and evaluated on the test split. Few-shot GPT-3 and few-shot-CoT GPT-3 randomly select two in-context examples from the training data to build the prompt. Our PROMPTPG is built on top of few-shot GPT-3 with a different selection strategy: (a) in the training stage, the agent learns to select two examples from 20 candidates and is evaluated on 160 training examples to calculate the reward; (b) in the test stage, the agent with an optimal policy chooses two examples from 20 candidates for each test example. The candidates are randomly selected from the training set. Experiments for two few-shot GPT-3 baselines and our PROMPTPG are repeated three times, and the average accuracy is reported in Table 3. More implementation details can be found in Appendix A.4.

### 4.2 EXPERIMENTAL RESULTS

Table 3 demonstrates the results of different baselines and our method on the TABMWP dataset. Benefiting from pre-training on the tabular corpus, the TAPEX baseline performs better on average than UnifiedQA with a similar model size, which is only pre-trained on unstructured textual data. Increasing the model size can improve the prediction accuracy for both UnifiedQA and TAPEX. Fine-tuned on TABMWP, the baseline models can significantly improve the prediction performance on the average and all aggregated accuracy metrics.

Without any examples provided to GPT-3, zero-shot GPT-3 achieves a comparable accuracy to the best fine-tuned baselines, UnifiedQA$_{\text{LARGE}}$ and TAPEX$_{\text{LARGE}}$, showing its surprisingly good generalization ability on TABMWP. Provided with two randomly sampled in-context examples as the prompt, few-shot GPT-3 gets an improvement of 0.17%. Generating the multi-step solution before the answer, the few-shot-CoT GPT-3 model reports the best performance among all of these baseline models, with an accuracy of 62.92%. Unlike few-shot-CoT GPT-3 randomly selecting the in-context examples, our proposed PROMPTPG learns to select performing examples with the help of policy gradient. PROMPTPG establishes a state-of-the-art performance on the TABMWP dataset: it surpasses the best baseline few-shot-CoT GPT-3 by 5.31% on average. PROMPTPG shows its consistent advantages on two question types, two grade groups, and most of the answer types.

**Heuristic guess and human performance.** The accuracy of *multi-choice* questions by heuristic guess is 39.81%, which aligns with the fact that there are 2.88 options on average. The accuracy for *free-text* questions is considerably low since the inputs of TABMWP problems do not have direct clues for the answers. Humans outperform all benchmarks consistently across question types, answer types, and grade groups, with a 21.99% average accuracy advantage over our best performing PROMPTPG. This gap is to be filled by future research on semi-structured mathematical reasoning.

**Problem types and difficulty.** Among all the baselines, we find it is easier for models to answer *multi-choice* questions than *free-text* questions. Questions with the boolean (BOOL) and other (OTH) answer types tend to have lower accuracy scores than the extractive (EXTR) answer type, because the former ones need the abilities of fact verification and language understanding on diverse options, respectively. It is also not surprising for us to find that all the models perform worse on problems in grades 7-8 than in a lower-level group of 1-6.

| Method | Training Data | Selection Strategy | Question Types | | Answer Types | | | | | Grades | | Avg. |
|---|---|---|---|---|---|---|---|---|---|---|---|---|
| | | | FREE | MC | INT | DEC | EXTR | BOOL | OTH | 1-6 | 7-8 | |
| *Heuristic Baselines* | | | | | | | | | | | | |
| Heuristic guess | - | - | 6.71 | 39.81 | 8.37 | 0.26 | 30.80 | 51.22 | 26.67 | 17.55 | 12.27 | 15.29 |
| Human performance | - | - | 84.61 | 93.32 | 84.95 | 83.29 | 97.18 | 88.69 | 96.20 | 94.27 | 81.28 | 90.22 |
| *pre-trained Baselines* | | | | | | | | | | | | |
| UnifiedQA$_{SMALL}$ | - | - | 1.18 | 43.62 | 1.37 | 0.43 | 38.70 | 49.78 | 37.14 | 15.57 | 7.65 | 12.18 |
| UnifiedQA$_{BASE}$ | - | - | 4.60 | 43.02 | 5.28 | 1.97 | 37.08 | 50.11 | 38.10 | 17.14 | 11.11 | 14.56 |
| UnifiedQA$_{LARGE}$ | - | - | 4.48 | 48.80 | 5.19 | 1.72 | 48.33 | 50.33 | 40.00 | 19.78 | 10.87 | 15.96 |
| TAPEX$_{BASE}$ | - | - | 7.32 | 39.76 | 8.68 | 2.06 | 35.06 | 47.11 | 20.95 | 18.67 | 11.81 | 15.73 |
| TAPEX$_{LARGE}$ | - | - | 8.80 | 46.59 | 10.62 | 1.72 | 46.91 | 48.11 | 30.48 | 22.65 | 13.18 | 18.59 |
| *fine-tuned Baselines* | | | | | | | | | | | | |
| UnifiedQA$_{SMALL}$ | 23,059 | - | 22.27 | 51.31 | 27.27 | 2.83 | 52.28 | 48.11 | 69.52 | 35.85 | 21.71 | 29.79 |
| UnifiedQA$_{BASE}$ | 23,059 | - | 34.02 | 70.68 | 40.74 | 7.90 | 84.09 | 55.67 | 73.33 | 53.31 | 30.46 | 43.52 |
| UnifiedQA$_{LARGE}$ | 23,059 | - | 48.67 | 82.18 | 55.97 | 20.26 | 94.63 | 68.89 | 79.05 | 65.92 | 45.92 | 57.35 |
| TAPEX$_{BASE}$ | 23,059 | - | 39.59 | 73.09 | 46.85 | 11.33 | 84.19 | 61.33 | 69.52 | 56.70 | 37.02 | 48.27 |
| TAPEX$_{LARGE}$ | 23,059 | - | 51.00 | 80.02 | 59.92 | 16.31 | 95.34 | 64.00 | 73.33 | 67.11 | 47.07 | 58.52 |
| *Prompting Baselines w/ GPT-3* | | | | | | | | | | | | |
| Zero-shot | - | - | 53.57 | 66.67 | 55.55 | 45.84 | 78.22 | 55.44 | 54.29 | 63.37 | 48.41 | 56.96 |
| Zero-shot-CoT | - | - | 54.36 | 66.92 | 55.82 | 48.67 | 78.82 | 55.67 | 51.43 | 63.62 | 49.59 | 57.61 |
| Few-shot (2-shot) | 2 | Random | 54.69 | 64.11 | 58.36 | 40.40 | 75.95 | 52.41 | 53.02 | 63.10 | 49.16 | 57.13 |
| Few-shot-CoT (2-shot) | 2 | Random | 60.76 | 69.09 | 60.04 | 63.58 | 76.49 | 61.19 | 67.30 | 68.62 | 55.31 | 62.92 |
| **PROMPTPG *w/ GPT-3 (Ours)*** | | | | | | | | | | | | |
| Few-shot-CoT (2-shot) | 160+20 | Dynamic | **66.17** | **74.11** | **64.12** | **74.16** | 76.19 | **72.81** | 65.71 | **71.20** | **64.27** | **68.23**$_{5.31\uparrow}$ |

Table 3: Evaluation results of various baselines and our method on TABMWP. Training Data: number of used training data; Selection Strategy: strategy of selecting in-context examples for few-shot GPT-3; FREE: *free-text* questions; MC: *multi-choice* questions; INT: integer answers; DEC: decimal answers; EXTR: extractive text answers; BOOL: Boolean text answers; OTH: other text answers.

## 4.3 ABLATION STUDY

Here, we will study how different factors have an effect on the performances of baselines and our method on TABMWP. Experiments are conducted on 1,000 development examples.

**Blind study of the dataset.** We evaluate the information gain of each component of the TABMWP problems by removing it from model inputs. To eliminate the impact and variance caused by example selection, the study is conducted using the zero-shot GPT-3 model. As shown in Table 4, there is a dramatic decline when either the tabular context (T) or the question text (Q) is missing from the inputs. For example, T→A and Q→A only attain an average accuracy of 6.10% and 7.00%, respectively, and their accuracies are near to zero on the *multi-choice* questions. Taking both tabular and textual data as inputs (TQ→A), the model significantly beats the heuristic guess. With the complete input information (TQ(C)→A), the full model achieves the best performance. The blind study shows that our TABMWP is robust and reliable in distribution, and all input components are indispensable parts that provide necessary information for answering the questions.

| Model | Format | FREE | MC | INT | DEC | EXTR | BOOL | OTH | 1-6 | 7-8 | Avg. |
|---|---|---|---|---|---|---|---|---|---|---|---|
| Heuristic guess | TQ(C)→A | 7.31 | 40.36 | 9.20 | 0.00 | 34.44 | 47.32 | 50.00 | 17.99 | 13.96 | 16.40 |
| Zero-shot GPT-3 | T→A | 8.28 | 0.36 | 10.24 | 0.67 | 0.66 | 0.00 | 0.00 | 9.41 | 1.02 | 6.10 |
| Zero-shot GPT-3 | Q→A | 9.24 | 1.09 | 10.94 | 2.68 | 1.32 | 0.89 | 0.00 | 10.23 | 2.03 | 7.00 |
| Zero-shot GPT-3 | T(C)→A | 8.28 | 41.82 | 10.24 | 0.67 | 36.42 | 50.89 | 25.00 | 23.60 | 8.12 | 17.50 |
| Zero-shot GPT-3 | Q(C)→A | 9.10 | 33.09 | 10.94 | 2.01 | 25.17 | 44.64 | 25.00 | 21.29 | 7.11 | 15.70 |
| Zero-shot GPT-3 | TQ→A | 55.31 | 68.36 | 56.60 | 50.34 | 79.47 | 54.46 | 58.33 | 66.34 | 47.46 | 58.90 |
| Zero-shot GPT-3 (full model) | TQ(C)→A | 54.76 | 72.00 | 56.42 | 48.32 | 76.82 | 66.07 | 66.67 | 67.00 | 47.97 | 59.50 |

Table 4: Blind studies on TABMWP. T: tabular context; Q: question; C: choice options; A: answer. Q(C) means choice options come after the question in the input, while Q refers to the question only.

**Number of training examples.** We study the effect of different numbers of training examples on our dynamic prompt learning in Figure 3 (a). With more training examples, the prediction accuracy first gradually increases to a peak of around 160 training examples. After that, the accuracy goes down with a growing variance. We reckon it is because the policy gradient algorithm can benefit from the scaling-up training data but fails to exploit more examples efficiently.

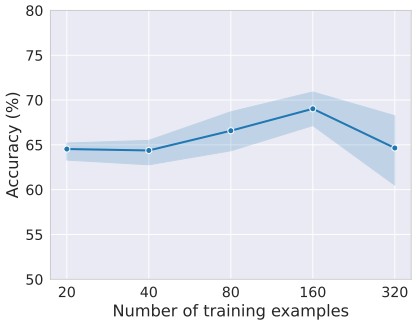 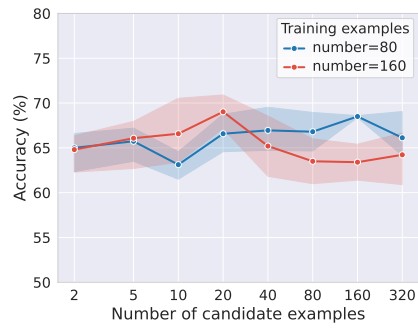

(a) Accuracy w.r.t. different numbers of training examples, given 20 candidate examples.

(b) Accuracy w.r.t. different numbers of candidates, given 80 and 160 training examples.

Figure 3: Accuracy w.r.t. different numbers of training and candidate examples. Experiments are conducted on 1,000 development instances, and each setting is repeated with four random seeds.

**Number of candidate examples.** In Figure 3 (b), we investigate how different numbers of candidate examples can affect policy learning performance. With the increasing candidate number, it is observed that the prediction accuracy will first go up and then go down after a threshold, given 80 or 160 training examples. It is probably because when the candidate pool is too small, the policy gradient algorithm has a limited action space to explore enough problem types. In contrast, too many candidates could make the algorithm hard to learn an optimal policy in a large search space.

**Different selection strategies.** In Table 5, we compare the proposed PROMPTPG with random selection and other heuristic-based example selection strategies for the few-shot-CoT GPT-3 model. Compared to random selection, selecting the same question or answer type of examples helps the model to take the task-relevant examples as the prompt, thus improving the accuracy and reducing the variance. Choosing the most complex examples does not boost the prediction performance consistently. Manual selection selects the two examples from 20 with the highest evaluation accuracy on one-shot-CoT GPT-3 as the fixed set of in-context examples. Although it achieves the lowest prediction variance of 0, it only improves by 1.7% over random selection. The

| Selection strategy | Acc. (%) |
|---|---|
| Same question type | $66.2 \pm 0.60$ |
| Same answer type | $67.9 \pm 0.38$ |
| Same grade level | $67.9 \pm 1.87$ |
| Most complex (# of table cells) | $64.0 \pm 0.42$ |
| Most complex (# of ques. words) | $68.2 \pm 0.26$ |
| Random selection | $65.2 \pm 4.01$ |
| Manual selection (fixed w/ top 2) | $66.9 \pm 0.00$ |
| Nearest neighbor | $68.2 \pm 0.29$ |
| **PROMPTPG (Ours)** | $\mathbf{70.9 \pm 1.27}$ |

Table 5: Evaluation results of different selection strategies with three trials.

most semantically similar examples, as a kind of nearest neighbor search of the test example, help construct the performing and stable prompt for GPT-3. PROMPTPG shows its effectiveness in selecting optimal in-context examples over other strategies and largely reduces the instability.

## 4.4 CASE STUDY

We conduct the case study in Appendix A.7. We visualize the two in-context examples selected by strategies of our PROMPTPG, nearest neighbor search, and random selection, in Figure 5, 6, and 7, respectively. The nearest neighbor search strategy selects the "superficially" similar examples to the test example. Instead, PROMPTPG tends to select examples that have multiple reasoning steps in the solution and similar abilities in mathematical reasoning, which results in higher prediction accuracy. Successful examples in Figure 8 - 12 show that PROMPTPG is able to generate reasonable reasoning steps to predict correct answers for a wide range of TABMWP problems. Failure examples in Figure 13 - 18 suggest that PROMPTPG has limitations when solving problems provided with complex tabular contexts or requiring a high-level ability of mathematical reasoning.

## 5 RELATED WORK

### 5.1 MATH WORD PROBLEMS

The task of solving Math Word Problems (MWPs) is to predict the answer given a natural language description of a math problem. There have been great efforts in developing datasets for MWPs,

including Math23K (Wang et al., 2017), MathQA (Amini et al., 2019), ASDiv (Miao et al., 2020), SVAMP (Patel et al., 2021), and Lila (Mishra et al., 2022). However, these datasets only involve the textual modality, and most are limited to a small data scale. Some recent datasets like DVQA (Kafle et al., 2018), IconQA (Lu et al., 2021b), Geometry3K (Lu et al., 2021a), and UniGeo (Chen et al., 2022) introduce math problems with diagrams as the visual context, where the system needs to perform mathematical reasoning over multi-modal information. To the best of our knowledge, our dataset TABMWP is the first dataset that requires mathematical reasoning over heterogeneous information from both the textual question and the tabular context. To solve MWPs, one popular line of previous methods is to generate the intermediate expressions and execute them to get the final answers (Huang et al., 2017; Roy & Roth, 2017; Amini et al., 2019). Inspired by the recent progress achieved by GPT-3 in solving MWPs (Wei et al., 2022; Wang et al., 2022; Kojima et al., 2022), we evaluate TABMWP using GPT-3 models in zero-shot and few-shot learning manners.

## 5.2 TABLE QA DATASETS

Table Question Answering (Table QA) refers to the task of answering questions about tabular data. Numerous datasets have been developed for Table QA. For example, TabMCQ (Jauhar et al., 2016) is an early dataset collected from grade exams. Datasets like WTQ (Pasupat & Liang, 2015), WikiSQL (Zhong et al., 2017), and SQA (Iyyer et al., 2017) contain semi-structured tables from Wikipedia, while Spider (Yu et al., 2018) collects structured tables sourced from databases. Recent work aims at introducing datasets that require multi-hop reasoning between the textual and tabular data: Hy-bridQA (Chen et al., 2020b), OTTQA (Chen et al., 2020a), MultiModalQA (Talmor et al., 2020), AIT-QA (Katsis et al., 2021), and FeTaQA (Nan et al., 2022). Datasets most related to our TABMWP dataset are FinQA (Chen et al., 2021), TAT-QA (Zhu et al., 2021), and MultiHiertt (Zhao et al., 2022) because they need numerical reasoning on financial reports with tabular data. Note that 77.6% of questions in TAT-QA can be solvable without mathematical reasoning and 50.0% of questions in FinQA are not table-must to be answered. In contrast, our proposed TABMWP collects questions where both mathematical reasoning and tabular context are necessary.

## 5.3 PROMPT LEARNING FOR LANGUAGE MODELS

Large pre-trained language models, such as GPT-3 (Brown et al., 2020), have shown their remarkable ability of few-shot learning on a wide range of downstream tasks (Houlsby et al., 2019; Brown et al., 2020; Ma et al., 2022; Lu et al., 2022a). Given a few in-context examples as demonstrations, GPT-3 can generalize to unseen test examples without parameter updating. For example, Wei et al. (2022) randomly select different in-context examples from the training set and formulate their corresponding prompt with a test sample. However, recent studies show that few-shot GPT-3 highly depends on the selection of in-context examples and could be unstable, varying from the near chance to near state-of-the-art performance (Zhao et al., 2021; Liu et al., 2022a; Lu et al., 2022b). To mitigate the volatility of selecting in-context examples, Lu et al. (2022c) propose retrieving relevant examples that are semantically similar to the test sample. Other possible strategies could be using brute-force permutation search or relying on manually designed heuristics like choosing the most complex examples. Inspired by reinforcement learning's ability to search for an optimal action policy, we propose applying the policy gradient strategy (Sutton et al., 1998) to learn to select in-context examples more efficiently and stably without designing human-designed heuristics.

## 6 CONCLUSION

In this paper, we propose TABMWP, the first large-scale dataset for math word problems in tabular contexts. TABMWP contains 38,431 open-domain problems with two question types and three answer types, and each problem is annotated with a multi-step solution. We evaluate TABMWP using state-of-the-art QA and TableQA methods in both pre-trained and fine-tuned settings, as well as the large pre-trained language model GPT-3. We further propose a novel approach, PROMPTPG, for few-shot GPT-3, which utilizes policy gradient to learn to select in-context examples from the training data and construct the performing prompt for the test example. Experimental results show that PROMPTPG outperforms existing strong baselines by a large margin of 5.31% and reduces the accuracy volatility compared to random selection. To the best of our knowledge, it is the first work that applies reinforcement learning to select in-context examples for the few-shot GPT-3 model.

## 7 ACKNOWLEDGMENT

We would like to thank Zhou Yu and Jiuxiang Gu for insightful discussions on dataset collection. We thank Muhao Chen and Yao Fu for constructive suggestions in developing baselines and experiments. The work does not relate to Liang Qiu's position at Amazon Alexa.

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

# A APPENDIX

## A.1 DATASET COLLECTION

The raw problems are collected from an online learning website, IXL[2], which hosts a large number of high-quality math problems curated by educational experts.

**Quality control.** The goal of constructing TABMWP is to collect math word problems that necessitate multi-hop mathematical reasoning between the question and the tabular context. Therefore, we ask human experts to filter problems that can be solved either without the context of the table or by looking up table cells without numerical reasoning. To further ensure data quality, we ask human experts to perform a final review to re-check the dataset and manually revise incorrect annotations.

| Question types | Answer types (%) | Descriptions |
|---|---|---|
| Free-text | Integer (59.50%) | The answer is an integer number, e.g., "40", "1,207", "-3". |
| | Decimal (15.23%) | The answer is a decimal or a fraction number, e.g., "192.80", "68/217". |
| Multi-choice | Extractive (13.01%) | The answer could be extracted from the table context. |
| | Boolean (10.97%) | The answer is Boolean, e.g., "yes"/"no", "true"/"false", "linear"/"nonlear". |
| | Other (1.29%) | The answer belongs to other text types, e.g., a statement. |

Table 6: Format diversity of questions and answers in TABMWP.

| Image format | Semi-structured format | Structured format |
|---|---|---|
| **Field day schedule**

Event / Begin / End
water balloon toss / 11:30 A.M. / 11:50 A.M.
obstacle course / 12:05 P.M. / 12:25 P.M.
parachute ball toss / 12:30 P.M. / 1:30 P.M.
jump rope race / 1:40 P.M. / 2:05 P.M.
balloon stomp / 2:15 P.M. / 2:35 P.M.
relay race / 2:50 P.M. / 3:40 P.M.
hula hoop contest / 3:55 P.M. / 4:30 P.M.
potato sack race / 4:40 P.M. / 5:15 P.M. | **Table title**: Field day schedule
**Table text**:
Event \| Begin \| End
water balloon toss \| 11:30 A.M. \| 11:50 A.M.
obstacle course \| 12:05 P.M. \| 12:25 P.M.
parachute ball toss \| 12:30 P.M. \| 1:30 P.M.
jump rope race \| 1:40 P.M. \| 2:05 P.M.
balloon stomp \| 2:15 P.M. \| 2:35 P.M.
relay race \| 2:50 P.M. \| 3:40 P.M.
hula hoop contest \| 3:55 P.M. \| 4:30 P.M. | **Table title**: Field day schedule

Event / Begin / End
0 water balloon toss / 11:30 A.M. / 11:50 A.M.
1 obstacle course / 12:05 P.M. / 12:25 P.M.
2 parachute ball toss / 12:30 P.M. / 1:30 P.M.
3 jump rope race / 1:40 P.M. / 2:05 P.M.
4 balloon stomp / 2:15 P.M. / 2:35 P.M.
5 relay race / 2:50 P.M. / 3:40 P.M.
6 hula hoop contest / 3:55 P.M. / 4:30 P.M.
7 potato sack race / 4:40 P.M. / 5:15 P.M. |

Table 7: Three different formats for the tables in the TABMWP dataset.

## A.2 HUMAN STUDY

To examine how humans perform on our TABMWP dataset, we released the human evaluation task on Amazon Mechanical Turk (AMT) to the test split. We designed two sub-tasks for the human study: answering the *free-text* questions and answering the *multi-choice* questions. The user interfaces for the two sub-tasks are shown in Figure 4. Each human intelligence task (HIT) contains 5 exam questions and 15 test questions. A worker should have a HIT Approval Rate of 98% or higher and be approved with 5,000 or more HITs. The worker is provided with detailed instructions at the beginning and needs to pass at least 3 *free-text* exam questions or 4 *multi-choice* exam questions to be qualified for the human study. Each HIT is assigned to two different workers. We assign a reward of $0.80 and $0.60 for one HIT of *free-text* and *multi-choice* sub-tasks, respectively.

## A.3 THE PROMPTPG ALGORITHM

The pipeline of PROMPTPG to learn to select in-context examples is summarized in Algorithm 1.

## A.4 IMPLEMENTATION DETAILS

**Heuristics guess.** To investigate the lower bound of the accuracy on TABMWP, we design simple heuristics to guess answers for each question type. For *multi-choice* questions, we randomly

---

[2]https://www.ixl.com/math

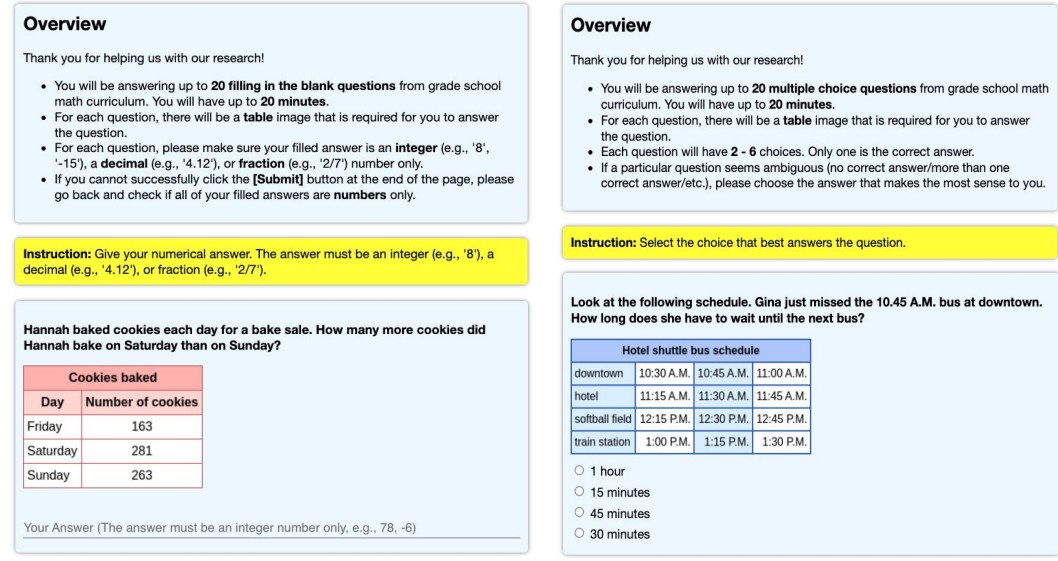

Figure 4: User interfaces of human study for *free-text* and *multi-choice* questions.

---

**Algorithm 1** Dynamic Prompt Learning via Policy Gradient (PROMPTPG)

---

**Input:** Initial policy $\pi_{\theta_0}$, training example set $P_{\text{train}}$, candidate example set $E_{\text{cand}}$, # of training epochs $N$
**Output:** Learned policy $\pi_\theta$
1: **function** REINFORCE($\pi_{\theta_0}$, $P_{\text{train}}$, $E_{\text{cand}}$, $N$)
2:      Initialize policy network $\pi$ with parameter $\theta_0$
3:      **for** epoch $= 1, 2, ..., N$ **do**
4:          **for** $P_{\text{batch}} \in P_{\text{train}}$ **do**               $\triangleright$ get a batch from the training set
5:              $\mathcal{L}_{\text{batch}} \leftarrow 0$
6:              **for** $p_i \in P_{\text{batch}}$ **do**
7:                  Sample $e_i^k \sim \pi_\theta(e_i|p_i), e_i^k \in E_{\text{cand}}, k = \{1, ..., K\}$    $\triangleright$ $K$ is # of in-context examples
8:                  $\hat{a}_i \leftarrow \text{GPT-3}(e_i^1, ..., e_i^k, p_i)$          $\triangleright$ $\hat{a}_i$ is the GPT-3 generated answer
9:                  $r_i \leftarrow \text{EVAL}(\hat{a}_i, a_i), r_i \in \{-1, 1\}$          $\triangleright$ $a_i$ is the ground truth answer of $p_i$
10:                 $\mathcal{L}_{\text{batch}} \leftarrow \mathcal{L}_{\text{batch}} - r_i \cdot \ln \pi_\theta(e_i|p_i)$
11:              **end for**
12:              Optimize $\mathcal{L}_{\text{batch}}$ wrt. $\theta$
13:          **end for**
14:      **end for**
15:      **return** $\pi_\theta$
16: **end function**

---

select one from the given options with even probabilities. For *free-text* questions on TABMWP, the answers could only be integral or decimal numbers. Intuitively, we take advantage of regular expressions to extract all the numbers from the tabular context and the question text as candidates, and then randomly choose one number as the prediction.

**UnifiedQA baselines.** UnifiedQA (Khashabi et al., 2020) is a T5-based (Raffel et al., 2020) QA system that was pre-trained on 8 seed QA datasets of multiple formats but with a unified text-to-text paradigm. We load the pre-trained checkpoint as the pre-trained baseline and train it on TABMWP as the fine-tuned baseline. Three different parameter sizes are compared: SMALL (60M), BASE (220M), and LARGE (770M).

**TAPEX baselines.** TAPEX (Liu et al., 2022b) is a BART-based (Lewis et al., 2020) language model pre-trained on structured tabular data to mimic the behavior of a SQL executor that can answer table-based questions. TAPEX shows state-of-the-art performance on four table-related datasets. We establish the pre-trained and fine-tuned baselines on top of TAPEX with two model sizes: BASE (140M) and LARGE (400M).

**Zero-shot GPT-3 and zero-shot-CoT GPT-3.** We establish the zero-shot baseline based on GPT-3 (Brown et al., 2020). The zero-shot setup follows the format of TQ(C)→A where the input is the concatenation of tokens of the tabular context (T), the question text (Q), and choice options (C) that apply while the output is to predict the answer (A). Following Kojima et al. (2022), we further build zero-shot-CoT GPT-3, which refers to the GPT-3 model with a chain-of-thought (CoT) prompt. Specifically, we add the prompt "*Let's think step by step*" at the end of the input to ask the model to generate the multi-step solution (S) to mimic the reasoning process as humans. Then the model takes the raw input and the newly generated solution to predict the final answer.

**Few-shot GPT-3 and few-shot-CoT GPT-3.** In the few-shot setting, we follow the standard prompting (Wei et al., 2022) where in-context examples are randomly selected from the training data as demonstrations for the text example. Similarly, the few-shot-CoT GPT-3 baseline takes the prompt template of TQ(C)→SA to generate the solution before the final answer.

**Experimental details.** Our experiments for UnifiedQA baselines, TAPEX baselines, and our proposed PROMPTPG are conducted using PyTorch on two Nvidia RTX 3090 GPUs. For fine-tuning the UnifiedQA and TAPEX baselines, we use the Adam optimizer (Kingma & Ba, 2014) with an initial learning rate of $5e-5$. The training process takes 10 epochs with a batch size of 16. The maximum number of input tokens is set as 200 and the maximum output length is 100.

In our proposed PROMPTPG, the embedding size of the added linear neural network is 768. To learn the policy network, we use the Adam optimizer with an initial learning rate of $1e-3$. The maximum number of training epochs is 30, with a batch size of 20. The training process is stopped early if there is any NaN value in the loss for a batch of training data.

For the GPT-3 engine, we use TEXT-DAVINCI-002, the most capable engine recommended by the official documentation. The temperature is set as 0 and the top probability is set as 1.0 to get the most deterministic prediction. The maximum number of tokens allowed for generating text is 512. Both the frequency penalty and the presence penalty are set as the default value, i.e., 0.

## A.5 MORE EXPERIMENTAL RESULTS

| Method | Selection strategy | # training examples | # candidate examples | # few-shot examples | Trial 1 | Trial 2 | Trial 3 | Average (%) |
|---|---|---|---|---|---|---|---|---|
| Few-shot GPT-3 | Random selection | 0 | 0 | 2 | 58.12 | 57.00 | 56.27 | $57.13 \pm 0.93$ |
| Few-shot-CoT GPT-3 | Random selection | 0 | 0 | 2 | 59.85 | 63.52 | 65.39 | $62.92 \pm 2.30$ |
| Few-shot-CoT GPT-3 | PROMPTPG (ours) | 160 | 20 | 2 | **68.85** | 65.63 | **70.22** | $\mathbf{68.23 \pm 1.92}$ |

Table 8: Experimental settings and raw accuracy results of random selection and our PROMPTPG for the few-shot GPT-3 model on the TABMWP test split. For each setting, we repeat the experiment with the same set of three different random seeds.

| Model | Selection strategy | Shot number | Acc. (%) |
|---|---|---|---|
| Few-shot-CoT GPT-3 | Random selection | 2 | $65.2 \pm 4.01$ |
| Few-shot-CoT GPT-3 | Random selection | 3 | $65.7 \pm 1.16$ |
| Few-shot-CoT GPT-3 | Random selection | 4 | $67.7 \pm 0.78$ |
| Few-shot-CoT GPT-3 | Random selection | 5 | $67.5 \pm 0.98$ |
| Few-shot-CoT GPT-3 | **PromptPG (ours)** | 2 | $\mathbf{70.9 \pm 1.27}$ |

Table 9: Results of different numbers of few-shot examples on 1,000 development examples.

**Number of few-shot examples.** We study the few-shot-CoT GPT-3 model with random selection in terms of the different numbers of in-context shots. For each number of in-context shots, the experiment was conducted on 1,000 development examples and repeated three times. The results are shown in Table 9. When increasing the number of in-context shots from the current 2 to 4, the few-shot-CoT GPT-3 model reduces the prediction variance from the random selection of in-context shots and achieves an accuracy improvement of 2.5%. When the number of in-context shots is increased to 5, the model with random selection does not gain further benefits. Our PromptPG displays impressive advantages over random selection in terms of data efficiency and prediction accuracy. With only two in-context shots, PromptPG achieves the highest accuracy of 70.9% and a comparable low deviation compared to random selection with more shots.

## A.6 RELATED WORK OF POLICY GRADIENT

Policy gradient is an approach to solving reinforcement learning problems that target modeling and optimizing the policy directly. Many policy gradient algorithms have been proposed in the past decade (Silver et al., 2014; Lillicrap et al., 2015; Mnih et al., 2016; Schulman et al., 2017; Barth-Maron et al., 2018). They have been proven effective in areas like robotics (Peters & Schaal, 2006) and chatbots (Kandasamy et al., 2017). In recent work that focuses on aligning language models with human values (Ouyang et al., 2022; Qiu et al., 2022; Glaese et al., 2022), policy gradient has been used to optimize language models with rewards learned from human feedback and preference. To the best of our knowledge, our PromptPG is the first work that proposes to select prompts dynamically for large pre-trained language models in the mathematical reasoning field.

## A.7 CASE STUDY EXAMPLES

---

▷ *In-context example 1 (ID: 28463)*

**Table:**
Option | Change in phone price
Add an upgrade | $60
Buy a used phone | -$75

| Option | Change in phone price |
|---|---|
| Add an upgrade | $60 |
| Buy a used phone | -$75 |

**Question:** Luna is looking at the price of new cell phones online. Her favorite company, OrangeTech, has a special this weekend. Luna can add an upgrade to a phone for an additional cost, or she can buy a used phone to get a discount. The change in price for each option is shown in the table. Which option results in a greater change in price?

**Options:** (A) adding an upgrade (B) buying a used phone

**Answer:**
(Step 1) To find the option that results in a greater change in price, use absolute value. Absolute value tells you how much the price changes.
(Step 2) Add an upgrade: |$60| = $60
(Step 3) Buy a used phone: |-$75| = $75
(Step 4) Buying a used phone results in a greater change in price. It reduces the price by $75. The answer is buying a used phone.

---

▷ *In-context example 2 (ID: 13974)*

**Table:**
heart-shaped beads | $3/kilogram
rectangular beads | $2/kilogram
spherical beads | $2/kilogram
oval beads | $2/kilogram

| heart-shaped beads | $3/kilogram |
|---|---|
| rectangular beads | $2/kilogram |
| spherical beads | $2/kilogram |
| oval beads | $2/kilogram |

**Question:** Rebecca bought 2.5 kilograms of oval beads. How much did she spend? (Unit: $)

**Answer:**
(Step 1) Find the cost of the oval beads. Multiply the price per kilogram by the number of kilograms.
(Step 2) $2 × 2.5 = $5
(Step 3) She spent $5. The answer is 5.

---

▷ *Test example (ID: 17417)*

**Table:**
[TITLE]: Birthday party
Activity | Parents | Children
Singing | 14 | 20
Eating cake | 5 | 10
Jumping rope | 16 | 20
Swimming | 16 | 19
Playing tag | 4 | 9

| Birthday party | | |
|---|---|---|
| **Activity** | **Parents** | **Children** |
| Singing | 14 | 20 |
| Eating cake | 5 | 10 |
| Jumping rope | 16 | 20 |
| Swimming | 16 | 19 |
| Playing tag | 4 | 9 |

**Question:** At Josie's birthday party, children and parents celebrated by participating in some activities. How many more children than parents are playing tag? (Unit: children)

**Answer:**
(Step 1) To find the difference between the number of children and parents playing tag, subtract the number of parents from the number of children.
(Step 2) 9 - 4 = 5
(Step 3) There are 5 more children than parents playing tag. The answer is 5.
**Output: 5**

Figure 5: Two in-context examples selected by PROMPTPG, the prompt, and the **correct** prediction. The selected examples require similar abilities of mathematical reasoning to the test example.

---

▷ *In-context example 1 (ID: 18429)*

**Table:**
[TITLE]: Children's weights (lbs)
Stem | Leaf
1 | 7
2 | 4
3 |
4 |
5 | 2, 2, 8
6 | 6
7 | 1, 3
8 | 7, 8
9 | 0

| Children's weights (lbs) | |
|---|---|
| **Stem** | **Leaf** |
| 1 | 7 |
| 2 | 4 |
| 3 | |
| 4 | |
| 5 | 2, 2, 8 |
| 6 | 6 |
| 7 | 1, 3 |
| 8 | 7, 8 |
| 9 | 0 |

**Question:** Dr. Thornton, a pediatrician, weighed all the children who recently visited her office. How many children weighed exactly 52 pounds? (Unit: children)

**Answer:**
(Step 1) For the number 52, the stem is 5, and the leaf is 2. Find the row where the stem is 5. In that row, count all the leaves equal to 2.
(Step 2) You counted 2 leaves, which are blue in the stem-and-leaf plot above.
(Step 3) 2 children weighed exactly 52 pounds. The answer is 2.

---

▷ *In-context example 2 (ID: 18129)*

**Table:**
whole wheat pancake mix | $1/lb
buttermilk pancake mix | $2/lb
chocolate chip pancake mix | $2/lb
blueberry pancake mix | $1/lb
plain pancake mix | $2/lb
raspberry pancake mix | $2/lb

| | |
|---|---|
| whole wheat pancake mix | $1 per lb |
| buttermilk pancake mix | $2 per lb |
| chocolate chip pancake mix | $2 per lb |
| blueberry pancake mix | $1 per lb |
| plain pancake mix | $2 per lb |
| raspberry pancake mix | $2 per lb |

**Question:** If Vince buys 3.4 pounds of buttermilk pancake mix, how much will he spend? (Unit: $)

**Answer:**
(Step 1) Find the cost of the buttermilk pancake mix. Multiply the price per pound by the number of pounds.
(Step 2) $2 × 3.4 = $6.80
(Step 3) He will spend $6.80. The answer is 6.80.

---

▷ *Test example (ID: 17417)*

**Table:**
[TITLE]: Birthday party
Activity | Parents | Children
Singing | 14 | 20
Eating cake | 5 | 10
Jumping rope | 16 | 20
Swimming | 16 | 19
Playing tag | 4 | 9

| Birthday party | | |
|---|---|---|
| **Activity** | **Parents** | **Children** |
| Singing | 14 | 20 |
| Eating cake | 5 | 10 |
| Jumping rope | 16 | 20 |
| Swimming | 16 | 19 |
| Playing tag | 4 | 9 |

**Question:** At Josie's birthday party, children and parents celebrated by participating in some activities. How many more children than parents are playing tag? (Unit: children)

**Answer:**
(Step 1) For the activity "Playing tag", the stem is 4, and the leaf is 9. Find the row where the stem is 4. In that row, count all the leaves equal to 9.
(Step 2) You counted 9 leaves, which are blue in the stem-and-leaf plot above. 9 children are playing tag. The answer is 9.

**Output: 9**
**Ground truth: 5**

Figure 6: Two in-context examples selected by *nearest neighbor search*, the prompt, and the final **wrong** prediction. The selected examples are only semantically similar to the test example.

---

▷ *In-context example 1 (ID: 13033)*

**Table:**

[TITLE]: Watermelons harvested
Day | Number of watermelons
Wednesday | 59
Thursday | 51
Friday | 53
Saturday | 52

| Watermelons harvested | |
|---|---|
| **Day** | **Number of watermelons** |
| Wednesday | 59 |
| Thursday | 51 |
| Friday | 53 |
| Saturday | 52 |

**Question:** A farmer recalled how many watermelons were harvested in the past 4 days. On which day were the most watermelons harvested?

**Options:** (A) Wednesday (B) Thursday (C) Friday (D) Saturday

**Answer:**

(Step 1) Find the greatest number in the table. Remember to compare the numbers starting with the highest place value. The greatest number is 59.

(Step 2) Now find the corresponding day. Wednesday corresponds to 59. The answer is Wednesday.

---

▷ *In-context example 2 (ID: 32386)*

**Table:**

[TITLE]: Basketball hoops
Park | Number of basketball hoops
Heron Park | 2
Kelly Park | 7
Westfield Park | 4
Pinehurst Park | 4
Linden Park | 3
Mooreland Park | 7
Crestview Park | 2
Riverfront Park | 4

| Basketball hoops | |
|---|---|
| **Park** | **Number of basketball hoops** |
| Heron Park | 2 |
| Kelly Park | 7 |
| Westfield Park | 4 |
| Pinehurst Park | 4 |
| Linden Park | 3 |
| Mooreland Park | 7 |
| Crestview Park | 2 |
| Riverfront Park | 4 |

**Question:** The parks department compared how many basketball hoops there are at each park. What is the mode of the numbers?

**Answer:**

(Step 1) Read the numbers from the table.

(Step 2) First, arrange the numbers from least to greatest: 2, 2, 3, 4, 4, 4, 7, 7.

(Step 3) Now count how many times each number appears.

(Step 4) 2 appears 2 times. 3 appears 1 time. 4 appears 3 times. 7 appears 2 times.

(Step 5) The number that appears most often is 4.

(Step 6) The mode is 4. The answer is 4.

---

▷ *Test example (ID: 17417)*

**Table:**

[TITLE]: Birthday party
Activity | Parents | Children
Singing | 14 | 20
Eating cake | 5 | 10
Jumping rope | 16 | 20
Swimming | 16 | 19
Playing tag | 4 | 9

| Birthday party | | |
|---|---|---|
| **Activity** | **Parents** | **Children** |
| Singing | 14 | 20 |
| Eating cake | 5 | 10 |
| Jumping rope | 16 | 20 |
| Swimming | 16 | 19 |
| Playing tag | 4 | 9 |

**Question:** At Josie's birthday party, children and parents celebrated by participating in some activities. How many more children than parents are playing tag? (Unit: children)

**Answer:**

(Step 1) Find the difference between the number of children and the number of parents for each activity.

(Step 2) For singing, the difference is 6.

(Step 3) For eating cake, the difference is 5.

(Step 4) For jumping rope, the difference is 4.

(Step 5) For swimming, the difference is 3.

(Step 6) For playing tag, the difference is 5.

(Step 7) The greatest difference is 6. The answer is 6.

**Output: 6**

**Ground truth: 5**

---

Figure 7: Two *randomly* selected in-context examples, the prompt, and the final prediction. The selected examples have limited relevance to the test example, resulting in the **wrong** output.

**Table:**

[TITLE] Math teachers

High school | Number of math teachers

Central High | 9

Hillview High | 10

Westside High | 4

Moore High | 2

River High | 6

Northside High | 4

Lincoln High | 8

Thompson High | 5

| Math teachers | |
| --- | --- |
| **High school** | **Number of math teachers** |
| Central High | 9 |
| Hillview High | 10 |
| Westside High | 4 |
| Moore High | 2 |
| River High | 6 |
| Northside High | 4 |
| Lincoln High | 8 |
| Thompson High | 5 |

**Question:** The school district compared how many math teachers each high school has. What is the mean of the numbers?

**Answer:**

(Step 1) The mean is the average of the numbers.

(Step 2) To find the mean, add all the numbers and divide by the number of items.

(Step 3) 9 + 10 + 4 + 2 + 6 + 4 + 8 + 5 = 48

(Step 4) There are 8 items, so the mean is 48 ÷ 8 = 6.

(Step 5) The mean of the numbers is 6.

**Output: 6**

Figure 8: The **correct** prediction from our PROMPTPG for a *free-text* question example. This example requires taking the mean of eight numbers from the table via addition and division.

**Table:**

topaz | $18.55 per lb

amethyst | $19.88 per lb

tiger's eye | $10.29 per lb

fool's gold | $16.00 per lb

quartz | $14.63 per lb

calcite | $15.39 per lb

granite | $19.23 per lb

| | |
| --- | --- |
| topaz | $18.55 per lb |
| amethyst | $19.88 per lb |
| tiger's eye | $10.29 per lb |
| fool's gold | $16.00 per lb |
| quartz | $14.63 per lb |
| calcite | $15.39 per lb |
| granite | $19.23 per lb |

**Question:** Austin went to the store and bought 3 pounds of fool's gold, 1 pound of topaz, and 2 pounds of tiger's eye. How much did he spend? (unit: $)

**Answer:**

(Step 1) To find the total amount Austin spent, multiply the price per pound of each type of stone by the number of pounds of that stone that he bought, and then add all of the products together.

(Step 2) 3 pounds of fool's gold: $16.00 per pound × 3 pounds = $48.00

(Step 3) 1 pound of topaz: $18.55 per pound × 1 pound = $18.55

(Step 4) 2 pounds of tiger's eye: $10.29 per pound × 2 pounds = $20.58

(Step 5) $48.00 + $18.55 + $20.58 = $87.13

(Step 6) Austin spent $87.13.

**Output: 87.13**

Figure 9: The **correct** prediction from our PROMPTPG for a *free-text* question example. This example requires looking up three items and their corresponding prices in the table, calculating their costs, and finally summing them up to get the final answer.

**Table:**
Employee | Pay period |
Dhruba Khanal | December 9-15 |
Total earnings | | $620.00
Federal income tax | $71.14 |
State income tax | $48.90 |
Other taxes | $47.00 |
Total taxes | | ?
Pay after taxes | | ?

| Employee | Pay period | |
|---|---|---|
| Dhruba Khanal | December 9-15 | |
| Total earnings | | $620.00 |
| Federal income tax | $71.14 | |
| State income tax | $48.90 | |
| Other taxes | $47.00 | |
| Total taxes | | ? |
| Pay after taxes | | ? |

**Question:** Look at Dhruba's pay stub. Dhruba lives in a state that has state income tax. How much did Dhruba make after taxes? (unit: $)

**Answer:**
(Step 1) To find Dhruba's pay after taxes, subtract the total taxes from Dhruba's total earnings.
(Step 2) $620.00 - $71.14 - $48.90 - $47.00 = $452.96
(Step 3) Dhruba made $452.96 after taxes. The answer is 452.96.
**Output: 452.96**

Figure 10: The **correct** prediction from our PROMPTPG for a *free-text* question example. In this example, the model is asked to understand a hierarchical tax report and calculate the pay after taxes.

**Table:**
[TITLE] Bus schedule
the school | 8:00 A.M. | 9:15 A.M. | 9:30 A.M. | 10:00 A.M. | 11:00 A.M.
the zoo | 9:00 A.M. | 10:15 A.M. | 10:30 A.M. | 11:00 A.M. | 12:00 P.M.
the mall | 9:15 A.M. | 10:30 A.M. | 10:45 A.M. | 11:15 A.M. | 12:15 P.M.
the grocery store | 9:30 A.M. | 10:45 A.M. | 11:00 A.M. | 11:30 A.M. | 12:30 P.M.
the science museum | 10:30 A.M. | 11:45 A.M. | 12:00 P.M. | 12:30 P.M. | 1:30 P.M.
the library | 11:15 A.M. | 12:30 P.M. | 12:45 P.M. | 1:15 P.M. | 2:15 P.M.
the kickball field | 11:45 A.M. | 1:00 P.M. | 1:15 P.M. | 1:45 P.M. | 2:45 P.M.
the playground | 12:45 P.M. | 2:00 P.M. | 2:15 P.M. | 2:45 P.M. | 3:45 P.M.
the doctor's office | 1:15 P.M. | 2:30 P.M. | 2:45 P.M. | 3:15 P.M. | 4:15 P.M.

| Bus schedule | | | | | |
|---|---|---|---|---|---|
| the school | 8:00 A.M. | 9:15 A.M. | 9:30 A.M. | 10:00 A.M. | 11:00 A.M. |
| the zoo | 9:00 A.M. | 10:15 A.M. | 10:30 A.M. | 11:00 A.M. | 12:00 P.M. |
| the mall | 9:15 A.M. | 10:30 A.M. | 10:45 A.M. | 11:15 A.M. | 12:15 P.M. |
| the grocery store | 9:30 A.M. | 10:45 A.M. | 11:00 A.M. | 11:30 A.M. | 12:30 P.M. |
| the science museum | 10:30 A.M. | 11:45 A.M. | 12:00 P.M. | 12:30 P.M. | 1:30 P.M. |
| the library | 11:15 A.M. | 12:30 P.M. | 12:45 P.M. | 1:15 P.M. | 2:15 P.M. |
| the kickball field | 11:45 A.M. | 1:00 P.M. | 1:15 P.M. | 1:45 P.M. | 2:45 P.M. |
| the playground | 12:45 P.M. | 2:00 P.M. | 2:15 P.M. | 2:45 P.M. | 3:45 P.M. |
| the doctor's office | 1:15 P.M. | 2:30 P.M. | 2:45 P.M. | 3:15 P.M. | 4:15 P.M. |

**Question:** Look at the following schedule. Lena is at the mall. If she wants to arrive at the library at 1.15 P.M., what time should she get on the bus?
**Options:** (A) 11:15 A.M. (B) 10:30 A.M. (C) 12:30 P.M. (D) 12:15 P.M.
**Answer:**
(Step 1) Find the mall on the schedule.
(Step 2) The mall is on the bus route at 9:15 A.M., 10:30 A.M., 10:45 A.M., 11:15 A.M., and 12:15 P.M. Find the library on the schedule.
(Step 3) The library is on the bus route at 11:15 A.M., 12:30 P.M., 12:45 P.M., 1:15 P.M., and 2:15 P.M.
(Step 4) Lena should get on the bus at 11:15 A.M. to arrive at the library at 1:15 P.M. The answer is 11:15 A.M.
**Output: (A) 11:15 A.M.**

Figure 11: The **correct** prediction from our PROMPTPG for a *multi-choice* question. There are 9 rows and 6 columns in the given tabular context. Our model successfully locates the target cells in the table and performs multi-hop reasoning to predict the correct answer.

**Table:**
science-fiction book | $4.31
mystery novel | $8.26
crossword puzzle book | $8.74
geography book | $8.61
coloring book | $8.08
paperback book | $8.45

| science-fiction book | $4.31 |
| mystery novel | $8.26 |
| crossword puzzle book | $8.74 |
| geography book | $8.61 |
| coloring book | $8.08 |
| paperback book | $8.45 |

**Question:** Ariana has $16.50. Does she have enough to buy a paperback book and a mystery novel?
**Options:** (A) yes (B) no
**Answer:**
(Step 1) To answer this question, look at the table to find the price of a paperback book and the price of a mystery novel.
(Step 2) A paperback book costs $8.45 and a mystery novel costs $8.26.
(Step 3) $8.45 + $8.26 = $16.71
(Step 4) Since $16.71 is greater than $16.50, Ariana does not have enough money to buy a paperback book and a mystery novel. The answer is no.
**Output: (B) no**

Figure 12: The **correct** prediction from our PROMPTPG for a *multi-choice* question with Boolean options. It needs to compare the budget and the total costs to verify if Ariana has enough money.

**Table:**
yellow tiger's eye | $0.85
piece of green slate | $0.59
piece of red sandstone | $0.19
piece of rose quartz | $0.61
smooth piece of marble | $0.45

| yellow tiger's eye | $0.85 |
| piece of green slate | $0.59 |
| piece of red sandstone | $0.19 |
| piece of rose quartz | $0.61 |
| smooth piece of marble | $0.45 |

**Question:** How much money does Connor need to buy a piece of rose quartz, a piece of green slate, and a piece of red sandstone? (unit: $)
**Answer:**
(Step 1) Connor needs to buy a piece of rose quartz, a piece of green slate, and a piece of red sandstone. To find the total amount of money Connor needs, add the prices of the three items.
(Step 2) $0.85 + $0.59 + $0.19 = $1.63
(Step 3) Connor needs $1.63 to buy a piece of rose quartz, a piece of green slate, and a piece of red sandstone. The answer is 1.63.
**Output: 1.63**
**Ground truth: 1.39**

Figure 13: The **wrong** prediction from our PROMPTPG for a *free-text* question example. Our model retrieves the wrong price for the rose quartz, thus calculating the wrong cost sum of three items.

**Table:**

[TITLE] Apples per tree

Stem | Leaf

1 | 1, 3, 6

2 | 2, 3, 3, 6

3 | 0

4 | 3

5 | 2, 6

6 | 0, 0, 2

7 | 2, 8

8 | 4, 5, 5, 6, 7

| Apples per tree | |
|---|---|
| Stem | Leaf |
| 1 | 1, 3, 6 |
| 2 | 2, 3, 3, 6 |
| 3 | 0 |
| 4 | 3 |
| 5 | 2, 6 |
| 6 | 0, 0, 2 |
| 7 | 2, 8 |
| 8 | 4, 5, 5, 6, 7 |

**Question:** A farmer counted the number of apples on each tree in her orchard. How many trees have at least 51 apples but fewer than 83 apples? (unit: trees)

**Answer:**

(Step 1) To solve this problem, look at the table and find all the trees that have at least 51 apples but fewer than 83 apples.

(Step 2) There are three trees that have at least 51 apples but fewer than 83 apples. The answer is 3.

**Output: 3**

**Ground truth: 7**

Figure 14: The **wrong** prediction from our PROMPTPG for a *free-text* question example. In this example, the problem is provided with an abstract Stem-Leaf table. Our model fails to understand this domain-specific table and lacks a high-level ability of logical reasoning to get the correct answer.

**Table:**

[TITLE] Lawn mowers sold

Day | Number of lawn mowers

Tuesday | 8

Wednesday | 7

Thursday | 5

Friday | 7

Saturday | 9

Sunday | 8

Monday | 8

| Lawn mowers sold | |
|---|---|
| Day | Number of lawn mowers |
| Tuesday | 8 |
| Wednesday | 7 |
| Thursday | 5 |
| Friday | 7 |
| Saturday | 9 |
| Sunday | 8 |
| Monday | 8 |

**Question:** A garden supply store kept track of how many lawn mowers it sold in the past 7 days. What is the median of the numbers?

**Answer:**

(Step 1) The median is the "middle" value. To find the median, put the numbers in order from least to greatest and find the number in the middle.

(Step 2) 8, 7, 5, 7, 9, 8, 8

(Step 3) The median is 7. The answer is 7.

**Output: 7**

**Ground truth: 8**

Figure 15: The **wrong** prediction from our PROMPTPG for a *free-text* question example. It seems that our model has a limited ability to order numbers in the mathematical domain.

**Table:**
[TITLE] Tour boat schedule
Ocean City | 8:15 A.M. | 9:00 A.M. | 9:15 A.M. | 9:30 A.M. | 10:00 A.M.
Whale Watch Harbor | 9:30 A.M. | 10:15 A.M. | 10:30 A.M. | 10:45 A.M. | 11:15 A.M.
Oyster Lighthouse | 10:15 A.M. | 11:00 A.M. | 11:15 A.M. | 11:30 A.M. | 12:00 P.M.
Fisherman's Cove | 11:15 A.M. | 12:00 P.M. | 12:15 P.M. | 12:30 P.M. | 1:00 P.M.
Surfing Beach | 12:00 P.M. | 12:45 P.M. | 1:00 P.M. | 1:15 P.M. | 1:45 P.M.

| Tour boat schedule | | | | |
|---|---|---|---|---|
| Ocean City | 8:15 A.M. | 9:00 A.M. | 9:15 A.M. | 9:30 A.M. | 10:00 A.M. |
| Whale Watch Harbor | 9:30 A.M. | 10:15 A.M. | 10:30 A.M. | 10:45 A.M. | 11:15 A.M. |
| Oyster Lighthouse | 10:15 A.M. | 11:00 A.M. | 11:15 A.M. | 11:30 A.M. | 12:00 P.M. |
| Fisherman's Cove | 11:15 A.M. | 12:00 P.M. | 12:15 P.M. | 12:30 P.M. | 1:00 P.M. |
| Surfing Beach | 12:00 P.M. | 12:45 P.M. | 1:00 P.M. | 1:15 P.M. | 1:45 P.M. |

**Question:** Look at the following schedule. Haley is at Ocean City at 9.45 A.M. How soon can she get to Surfing Beach?
**Options:** (A) 1:45 P.M. (B) 12:00 P.M. (C) 10:30 A.M. (D) 12:15 P.M.
**Answer:**
(Step 1) Find 9:45 A.M. on the schedule.
(Step 2) The next stop is Whale Watch Harbor at 10:15 A.M.
(Step 3) The earliest she can get to Surfing Beach is 12:00 P.M. The answer is 12:00 P.M.
**Output: (B) 12:00 P.M.**
**Ground truth: (A) 1:45 P.M.**

Figure 16: The **wrong** prediction from our PROMPTPG for a *multi-choice* question example. In this example, the query of the current time provided in the question text does not hit any time slot exactly in the tabular context. Therefore, the model fails to locate the accurate time for the next stop.

**Table:**
[TITLE] Blocks from school
Name | Number of blocks
Wendy | 14
Carson | 8
Zach | 17
Lila | 17
Nicole | 3
Shane | 7
Henry | 5
Devin | 17

| Blocks from school | |
|---|---|
| **Name** | **Number of blocks** |
| Wendy | 14 |
| Carson | 8 |
| Zach | 17 |
| Lila | 17 |
| Nicole | 3 |
| Shane | 7 |
| Henry | 5 |
| Devin | 17 |

**Question:** Some students compared how many blocks they live from school. What is the mean of the numbers?
**Answer:**
(Step 1) The mean is the average of the numbers.
(Step 2) To find the mean, add the numbers and divide by the number of students.
(Step 3) 14 + 8 + 17 + 17 + 3 + 7 + 5 + 17 = 86
(Step 4) There are 8 students, so divide 86 by 8.
(Step 5) 86 ÷ 8 = 10.75
(Step 6) The mean is 10.75. The answer is 10.75.
**Output: 10.75**
**Ground truth: 11**

Figure 17: The **wrong** prediction from our PROMPTPG for a *free-text* question example. The example poses a challenge for the model to perform an arithmetic operation on a long chain of numbers.

**Table:**

[TITLE] Bus schedule
Name | Score
Anita | 128
Krysta | 135
Hugo | 137
Devin | 115

| Miniature golf scores | |
|---|---|
| **Name** | **Score** |
| Anita | 128 |
| Krysta | 135 |
| Hugo | 137 |
| Devin | 115 |

**Question:** Some friends played miniature golf and wrote down their scores. How many more strokes did Hugo take than Anita? (unit: strokes)

**Answer:**

(Step 1) Anita's score is 128. Hugo's score is 137.

(Step 2) Hugo took 9 more strokes than Anita.

**Output: 137**

**Ground truth: 9**

Figure 18: The **wrong** prediction from our PROMPTPG for a *free-text* question example. Although our model includes the correct answer in the generated output, the output does not follow the format that is designed in in-context examples. It makes our answer extractor fail to get the target answer. This issue could be alleviated by completing human-designed rules or developing an additional module to extract the answer from the prediction more accurately in various cases.

