# OpenReview forum: "Dynamic Prompt Learning via Policy Gradient for Semi-structured Mathematical Reasoning"
_ICLR.cc/2023/Conference — ICLR 2023 poster_

### Official Review · Reviewer_MEHB · 2022-10-22

**Confidence:** 3
**Correctness:** 3
**Technical Novelty And Significance:** 3
**Empirical Novelty And Significance:** 3
**Recommendation:** 5

**Clarity, Quality, Novelty And Reproducibility:**

This work proposed a novel and high-quality dataset TABMWP for math word problems with tabular context, which could be used as a benchmark dataset of tabular reasoning domains. The presentation of the proposed approach is not very clear, which makes it hard to evaluate its originality and the key parts that improve over other baselines.

**Strength And Weaknesses:**

Strength:
-

[+] The proposed dataset is novel and of high quality.

[+] The proposed approach, built on top of [1] and [2], applies reinforcement learning to select in-context examples for the few-shot GPT-3 model and has better performance over state-of-the-art QA, and TableQA methods in most settings.

[+] The proposed approach also has better performance than some other selection strategies of in-context examples.

Weakness:
-

[-] The authors should investigate the performance gain of PromptPG over prompting baselines w/ GPT-3 in relation to the usage of many times more training examples.

[-] The presentation of this work misses some important details of the proposed approach PromptPG (see below).

A few detailed questions related to the proposed approach PromptPG:

1. What is the structure of the prompt creator in Figure 2? Does it contain learnable parameters?

2. How does PromptPG handle the structured tabular data exactly? Are there steps for generating queries to the tabular data? If so, where are the steps in Figure 2?

3. How do you select the candidate examples? Are they randomly drawn from the training data? How would different candidate pools affect the performance of selecting in-context examples?

---

[1] Jiachang Liu, Dinghan Shen, Yizhe Zhang, William B Dolan, Lawrence Carin, and Weizhu Chen.
What makes good in-context examples for GPT-3? In Proceedings of Deep Learning Inside Out
(DeeLIO 2022): The 3rd Workshop on Knowledge Extraction and Integration for Deep Learning
Architectures, pp. 100–114, 2022a.

[2] Qian Liu, Bei Chen, Jiaqi Guo, Morteza Ziyadi, Zeqi Lin, Weizhu Chen, and Jian-Guang Lou.
Tapex: Table pre-training via learning a neural SQL executor. In International Conference on
Learning Representations (ICLR), 2022b.

**Summary Of The Paper:**

This work proposed a novel dataset, TABMWP, for math word problems with tabular context, which contains over 30,000 open-domain problems that require multiple steps of reasoning. On top of the dataset, it further proposed a new approach to learning to select in-context examples and construct the performing prompt for the test example. It then conducted extensive experiments evaluating the performance of the proposed approach, GPT-3, state-of-the-art QA, and TableQA methods on the new dataset TABMWP. The experiment results demonstrate the better performance of PromptPG in many cases.

**Summary Of The Review:**

This work proposed a novel and high-quality dataset TABMWP for math word problems with tabular context, which could be used as a benchmark dataset of tabular reasoning domains. It then proposed a new approach PromptPG which applies reinforcement learning to select in-context examples for the few-shot GPT-3. The work also conducts extensive experiments to evaluate the proposed approach over many other baselines.

---

> ### Author Response · Authors · 2022-11-11
> **Response to Reviewer MEHB (R5)**
>
> Dear reviewer, thank you for the insightful comments, and we appreciate your time and effort. We are excited that you think our dataset is novel and of high quality. We are glad that you recognize that our approach, PromptPG, has better performance over SOTA QA and Table QA methods and has better performance than other selection strategies for in-context examples. We are willing to address your concerns below.
>
> If you have any further questions, please feel free to let us know. We are grateful to have the chance to resolve these questions with you in the discussion phase!
>
> > **Q1: PromptPG over prompting baselines w/ GPT-3 with more training examples.**
>
> Thank you for your suggestion! In the submission paper, we compared PromptPG with more training examples in the ablation study and reported the results in Figure 3 (a). With more training examples, the prediction accuracy increases to a peak of around 160 training examples. After that, the accuracy goes down with a growing variance. Therefore, our final PromptPG takes 160 training examples, and we report its result on the test set in Table 3.
>
> > **Q2: What is the structure of the prompt creator in Figure 2? Does it contain learnable parameters?**
>
> We introduced how to create a prompt in Section 3.1. Specifically, a prompt consists of a few training examples and a test example. Each training example consists of a table context $t$, a question $q$, options $c$ that apply, and an answer $a$. Following Wei et al. (2022), a solution $s$ can be augmented in front of the answer $a$ to reveal the multi-step reasoning process. The test example has a similar format, except that it does not contain the answer and solution. The GPT-3 model takes the prompt as the input and generates the solution as the output. We visualize the examples of the prompts and the predictions for our PromptPG, nearest neighbor search, and random selection in Figures 16, 17, and 18, respectively, in the Appendix.
>
> No, the prompt creator does not contain learnable parameters. We followed chain-of-thought prompting (Wei et al., 2022) to construct the prompt based on templates.
>
> > **Q3: How does PromptPG handle the structured tabular data exactly?**
>
> No. As we claimed in Section 2.1, "we focus on the semi-structured format as the table context for simplicity". Our PromptPG takes the semi-structured format as the tabular context and does not use structured tabular data. Other GPT-3 baselines and the UnifiedQA baselines use the semi-structured format as well. The structured tabular data is represented as a structured spreadsheet in TabMWP and is only used for SQL-based methods like TAPEX. More details regarding the tabular formats can be found in Section 2.2 and Appendix A.2.
>
> > **Q4: How do you select the candidate examples?**
>
> As the implementation details explain in Section 4.2, "the candidates are randomly selected from the training set". For each trial, we keep the same random seed fixed when selecting the candidate examples from the training set for each test problem.
>
> > **Q5: How would different candidate pools affect the performance of selecting in-context examples?**
>
> In the submitted paper, we have compared the impact of different candidate pools on PromptPG in terms of different numbers of candidates and different candidate pools. In Figure 3 (b), we investigated how different numbers of candidate examples can affect policy learning performance. We can see that with the increasing candidate number, the prediction accuracy will first go up and then go down after a threshold. Our PromptPG achieves optimal performance with 20 candidates, given 160 training examples. In Table 8 in the Appendix, we examined how different candidate pools affect the performance of selecting in-context examples with different seeds. The few-shot-CoT GPT-3 model with PromptPG has a lower variance and a higher average accuracy than the model with random selection.
>
> > **Q6: Two mentioned papers in the comment.**
>
> Dear reviewer, would you mind giving us more information about your comments on these two papers? We already cited these two papers in the paper and implemented two baselines on top of them. We chose TAPEX [2] as one of the SOTA TableQA methods and evaluated TAPEX in both pre-trained and fine-tuning settings with two model sizes on TabMWP. The results are reported in Table 3. To select in-context examples, paper [1] proposes a nearest neighbor search strategy. We have compared different selection strategies, including nearest neighbor search, in Table 5.

---

> ### Author Response · Authors · 2022-12-05
> **Looking forward to a discussion before the deadline**
>
> Dear Reviewer,
>
> Thanks again for your great efforts in reviewing our paper!
>
> We have addressed all your questions in detail. As the deadline for the discussion is fast approaching, we are really looking forward to having a discussion with you on the OpenReview system. Would you mind checking our response and letting us know if you have further questions?
>
> With sincere regards,
>
> Authors of Paper 828

---

### Official Review · Reviewer_5Mut · 2022-10-24

**Confidence:** 3
**Correctness:** 2
**Technical Novelty And Significance:** 3
**Empirical Novelty And Significance:** 3
**Recommendation:** 6

**Clarity, Quality, Novelty And Reproducibility:**

The TABMWP dataset is helpful for researches which use general-purposed LM to perform mathematical reasoning tasks.
The idea of using reinforcement learning policy gradient strategy to train an agent to find helpful prompting examples is interesting.

**Strength And Weaknesses:**

Strength:
a)	The idea of using reinforcement learning policy gradient strategy to train an agent to find helpful prompting examples is interesting. It enables “automatic prompt-tuning” to some extent, since directly tuning GPT-3 is hard.
b)	The author provided a math problem dataset annotated with quality text solutions. It is helpful for researches which use general-purposed LM to perform mathematical reasoning tasks (e.g. “chain-of-thought”).
c)	The author conducted a rich set of experiments to test the method and incorporated human performance.

Weakness:
a)	The idea of constructing a tabular math problem dataset is not very inspiring.
b)	Some statements are contradictory. In the introduction, the author claimed that retrieving semantically similar examples might not work well. But in the ablation study part, the result indicates that selecting similar examples as prompt does make an improvement.
c)	PROMPTPG needs to be compared with manually chosen and fixed in-context example. If one randomly chooses example for every test-time problem, obviously it can increase the variance of accuracy.
d)	Dynamically selecting prompt examples still might not be enough. It relies on quality human annotated solutions (when “chain-of-thought” is needed). Sometimes the effort to obtain such annotations is non-trivial.
e)	Though ablation study and case study were performed, there is no in-depth analysis of them.

**Summary Of The Paper:**

This paper presents TABMWP, a dataset of grade-school-level math problems with tabular context and human annotated solutions in text form. Then the author proposes a method called PROMPTPG to mitigate the unstable issue occurred when solving problems above in few-shot setting. The problems and solutions in dataset were originally collected from a math learning website. Then the author manually filtered them to ensure there’s no problem can be solved without the tabular context and numerical reasoning process and checked the solutions’ correctness. The idea of the method proposed is trying to find the most appropriate in-context examples for a given problem in test time. The policy to choose in-context example is learnt by a reinforcement learning policy gradient strategy.

**Summary Of The Review:**

The main problem of this paper is that the experiments do not validate the motivation of the paper.
(1) Is the selection of examples important for the TABMWP dataset dataset? PROMPTPG needs to be compared with manually chosen and fixed in-context example.
(2) The policy gradient method designed in this paper is independent of the tabular data. Therefore, why this method is proposed for tabular data is not explained. Although the authors mentioned in Section 3.2 that "may be more severe on TABMWP", this has not been verified.

---

> ### Author Response · Authors · 2022-11-11
> **Response to Reviewer 5Mut (R4) (Cont.)**
>
> > **Q6: How do experiments validate the motivation of the paper? Is the selection of examples important for the TABMWP dataset?**
>
> In the paper, we claim that *the few-shot GPT-3 model highly depends on the selection of in-context examples and it could be worse on TabMWP*. We validate this claim by reporting the raw results with the same set of three different random seeds in Table 8. For the few-shot GPT-3 model with random selection, the prediction accuracy could be as low as 56.27%, which is even worse than the zero-shot GPT-3 model and fully supervised UnifiedQA and TAPEX baselines. Using chain-of-thought prompting, the prediction accuracy of the few-shot GPT-3 with random selection ranges from 59.85% to 65.39%. The large accuracy variance indicates that few-shot GPT-3 is sensitive to the selection of in-context examples.
>
> We also claim in the paper that *our PromptPG could learn to select good in-context examples for the test example, resulting in lower prediction variance and higher accuracy than random selection*. This claim is thoroughly validated by the experimental results in Tables 3, 5, and 8. For example, the three trials of PromptPG achieve an accuracy of 68.85%, 65.63%, and 70.22%, respectively. As we can see, even the lowest accuracy of PromptPG in three trials (65.63%) is higher than the greatest accuracy of random selection (65.39%).
>
> We copied the results in Table 8 as follows:
>
> | Method | Selection strategy | Trial 1 | Trial 2 | Trial 3 |  Average (%) |
> | :---------: | :---------: | :---------: | :---------: | :---------: | :---------: |
> | Few-shot GPT-3 |  Random selection | 58.12 | 57.00 | 56.27 | 57.13 ± 0.93 |
> | Few-shot-CoT GPT-3 |  Random selection | 59.85 | 63.52 | 65.39 | 62.92 ± 2.30 |
> | Few-shot-CoT GPT-3 |  **PromptPG (ours)** | **68.85** | **65.63** | **70.22** | **68.23 ± 1.92** |
>
>
> > **Q7: Explain why the policy gradient method is proposed for tabular data.**
>
> Recent work has shown that the few-shot GPT-3 model is able to achieve remarkable performance on downstream tasks like solving math word problems, but its performance is unstable and highly relies on the selection of in-context examples. The instability issue could be worse on TabMWP where the problems feature diversity in terms of question types, answer types, domains, grades, and required reasoning abilities. Some recent attempts have been proposed to address this issue. For example, the nearest neighbor search strategy selects semantically similar examples and alleviates this issue with reduced prediction variance and improved accuracy to some extent. Other techniques could include designing human-designed rules, such as selecting the most complex examples as the prompt.
>
> Our motivation in this paper is to design an effective algorithm capable of automatically selecting good in-context examples for every test problem. And we take the TabMWP as a case-study benchmark to evaluate the proposed algorithm.
>
> Inspired by the success of reinforcement learning in solving planning problems, we propose a policy gradient method, PromptPG, that dynamically learns to select good in-context examples when interacting with the GPT-3 API. PromptPG demonstrates its impressive performance with only 20 candidates, 160 training examples, and no human-designed heuristics. A comprehensive experimental comparison with existing methods and selection strategies verifies the impressive performance of PromptPG. We believe it could be used as a plug-and-play module for a wide range of LLM-powered tasks.

---

> > ### Comment · Reviewer_5Mut · 2022-12-08
> > **response**
> >
> > Thank you for your response. Some of my concerns were addressed.

---

> > > ### Author Response · Authors · 2022-12-09
> > > **Eager to improve the paper during the discussion process**
> > >
> > > Dear esteemed reviewer,
> > >
> > > We truly appreciate the time and effort you have invested in reviewing our paper. We are delighted that some of your concerns have been addressed, and we are eager to continue working on improving the paper during the discussion process.
> > >
> > > If you have any additional comments or concerns, we would be grateful if you could kindly let us know. We are committed to resolving any issues in the best way possible.
> > >
> > > Wishing you all the best,
> > >
> > > The authors

---

> ### Author Response · Authors · 2022-11-11
> **Response to Reviewer 5Mut (R4)  (Cont.)**
>
> > **Q3a: Suggestion: PromptPG needs to be compared with manually chosen and fixed in-context examples.**
>
> Thank you for your insightful suggestion! We achieved a new selection strategy, termed **manual selection**, which selects the two best examples from 20 as the fixed set of in-context examples. To be specific, we first randomly select 20 in-context examples from the training data. We evaluate each in-context example based on its one-shot-CoT performance on GPT-3 on 1,000 development examples. We choose the top two in-context examples with the highest evaluation accuracy and use them as the fixed set of in-context examples for the GPT-3 model. The new result is added in Table 5 of the revised version.
>
> We copied part of the results in Table 5 below:
>
> | Selection strategy | Acc. (%) |
> | :----------------: | :---------: |
> | Random selection | 65.2 ± 4.01 |
> | Manual selection | 66.9 ± 0.00 |
> | Nearest neighbor | 68.2 ± 0.29 |
> | **PromptPG (ours)** | **70.9 ± 1.27** |
>
> Manual selection can achieve the lowest prediction variance of 0 by selecting the two best examples from a fixed set of in-context examples. However, it only gains an improvement of 1.7% over random selection. Instead, our PromptPG learns to select instance-relevant in-context examples for each test example from a small set of 20 candidates and is able to achieve an improvement of 5.7% over random selection.
>
>
> > **Q3b: If one randomly chooses an example for every test-time problem, obviously it can increase the variance of accuracy.**
>
> We have to clarify that, in each trial for random selection, we keep the random seed fixed when selecting two examples from the training set for each test-time problem. It means that we use **the same randomly selected examples** for every test-time problem. The variance of accuracy mainly comes from the the sensitivity nature of GPT-3 when given different prompts.
>
> > **Q4: Dynamically selecting prompt examples still might not be enough. Sometimes the effort to obtain such annotations is non-trivial.**
>
> Thank you for your comments!
>
> Our PromptPG for dynamically selecting prompt examples shows impressive performance improvements over random selection. Using 20 candidate examples annotated with answers and solutions and 160 training examples with ground truth answers only, PromptPG archives an accuracy of 68.23%, which outperforms current SOTA methods.
>
> PromptPG also outperforms fully supervised methods in terms of data usage efficiency and computing economics. On TabMWP, the SOTA approach TAPEX$_large$ is fine-tuned on 23,059 training examples and evaluated on 7,686 development examples. PromptPG, on the other hand, uses only 0.5% of the annotated data (20+160 examples) and outperforms the best fully supervised method, TAPEX$_large$, without any parameter updates.
>
> In real-world applications, the effort to obtain the required annotations for PromptPG is very limited. The developer can easily annotate 20 training examples with detailed solutions and 160 training examples with only ground truth answers. Also, there are no workloads for setting up an annotation page and doing a quality check of the annotations, which are critical in fully supervised settings.
>
> Thank you for your insightful comments! We do agree with you that we could further improve PromptPG. For example, in future work, we are considering extending our method to not only automatically select the in-context examples but also edit the created prompt.
>
> > **Q5: The in-depth analysis of the ablation study and case study.**
>
> In the ablation study, 1) we investigate the blind study of the dataset to verify if each input component in TabMWP is indispensable and provides necessary information for answering the questions; 2) we study the effect of different numbers of training examples on our dynamic prompt learning; 3) we research how different numbers of candidate examples can affect the policy learning performance; 4) we compare eight different selection strategies with our PromptPG and verify the effectiveness of PromptPG in selecting shot examples.
>
> In the case study, we visualize the in-context examples and predictions by PromptPG, nearest neighbor search, and random selection. We analyze the differences among these three selection strategies. We further visualize five successful examples and six failed examples by PromptPG, accompanied by a detailed analysis.

---

> ### Author Response · Authors · 2022-11-11
> **Response to Reviewer 5Mut (R4)**
>
> Dear reviewer, thank you for the insightful comments. We appreciate your time and effort. We are pleased that you consider our dataset to be of high quality and useful for researchers using general-purpose LM to perform mathematical reasoning tasks. We are encouraged that you noted that our method for finding helpful prompting examples is interesting, and we conducted a rich set of experiments to verify its effectiveness.
>
> We address your questions below. We are eager to hear your feedback on our comments in order to improve our paper.
>
> > **Q1: The idea of constructing a tabular math problem dataset.**
>
> To fill the gaps in existing datasets on math word problems and Table QA, we propose Tabular Math Word Problems (TabMWP), a new large-scale dataset that contains 38,431 math word problems with tabular context. To the best of our knowledge, TabMWP is the first dataset for math word problems in a tabular context.
>
> TabMWP contains diversity in terms of question types, answer types, and context formats. Each problem is annotated with a detailed solution that reveals the multi-step reasoning steps to ensure full explainability. To solve problems in TabMWP, a system requires multi-hop mathematical reasoning over heterogeneous information by looking up table cells given textual clues and conducting multi-step operations to predict the final answer.
>
> TabMWP differs from related datasets in various aspects, including: (1) TabMWP is the first dataset to study math word problems over tabular context on open domains and is the largest in terms of data size; (2) Problems in TabMWP are annotated with the tabular context, unlike previous MWP datasets. The tabular context contains structured information and requires a system to make multi-hop reasoning over heterogeneous information; (3) Unlike Table QA datasets like FinQA, TAT-QA, and MultiHiertt, a lack of either mathematical reasoning or the tabular context renders the problems in TabMWP unanswerable; (4) TabMWP contains 38,431 different problems, 37,644 different tables, 28,876 different questions, and 6,153 different answers. It covers open domains, three tabular formats, two question types, five answer types, and grades 1-8. Therefore, TabMWP has unique advantages in terms of diversity over existing datasets; (5) Each problem in TabMWP is annotated with a high-quality natural language solution to reveal the reasoning process. The solution annotations are very useful to develop powerful large language models for the TabMWP task; (6) Experiments on SOTA QA and Table QA models show that TabMWP poses new challenges to current advanced question answering and machine reasoning methods.
>
> Overall, TabMWP is a novel, well-designed, and high-quality dataset for math word problems and Table QA. As committed by Reviewers R1-fX43, R3-G2Gx, and R5-MEHB, TabMWP is a valuable resource and contribution to the community. We are releasing the dataset and codes and will work with the community to push related research forward on NLP and deep learning.
>
>
> > **Q2: Clarification: the claim regarding retrieving semantically similar examples might not work well on TabMWP.**
>
> Sorry for the confusion. Retrieving semantically similar examples, for example, retrieving the nearest neighbor for the test example, could reduce the prediction variance and improve the accuracy, as reported in Table 5. Based on the case study in Section 4.4, we discover that the nearest neighbor search strategy prefers semantically or "superficially" similar examples to the test example. The selected in-context examples might not have similar abilities in mathematical reasoning to the test example, which results in prediction failure.
>
> Take Figure 6 as an example. Because we used Bert-based sentence similarity as the metric to find semantically similar examples, the selected two in-context examples with words in the questions "How many children..." and "how much..." were similar to the test example with the words "How many more children...". However, the reasoning abilities revealed in the solutions of the in-context examples are different from those in the test example, which misleads the prediction and results in a wrong prediction.
>
> Thank you for pointing this out! We have clarified the statements mentioned in the revised paper.

---

> ### Author Response · Authors · 2022-12-01
> **Thanks to Reviewer 5Mut**
>
> Dear reviewer,
>
> Please accept our sincere thanks again for all your suggestions on our work. We greatly appreciate your time and great efforts in improving our paper! We hope our responses have answered your questions. We are happy to answer any questions you may have later.
>
> Best regards,
>
> Authors of paper 828

---

> ### Author Response · Authors · 2022-12-05
> **Looking forward to a discussion before the deadline**
>
> Dear Reviewer,
>
> Thanks again for your great efforts in reviewing our paper!
>
> We have addressed all your questions in detail. As the deadline for the discussion is fast approaching, we are really looking forward to having a discussion with you on the OpenReview system. Would you mind checking our response and letting us know if you have further questions?
>
> With sincere regards,
>
> Authors of Paper 828

---

### Official Review · Reviewer_G2Gx · 2022-10-25

**Confidence:** 4
**Correctness:** 4
**Technical Novelty And Significance:** 4
**Empirical Novelty And Significance:** 3
**Recommendation:** 8

**Clarity, Quality, Novelty And Reproducibility:**

- The paper is beautifully written and easy to follow, with effective figures, summaries, and comparisons to existing work.

- The dataset appears to be highly original (first of its kind), and the proposed promptPG method appears to be novel to the best of my knowledge.


**Strength And Weaknesses:**

### Strength:

- This paper introduces a well-designed math word dataset that contains tabular information, upon which the answers depend. The dataset contains diversity in terms of difficulty level (from different grades), answer format (free-text or multiple choice) and table format (structured, semi-structured, and image), along with the associated answer rationales. This is extremely valuable to the community for evaluating the performance of models when working with tabular datasets.

- The dynamic prompting method via policy gradient appears novel and interesting, and has been shown to improve performance over a number of baseline strategies.

- The experiments are well-designed and covers a broad range of ablations and a large set of baselines, including human performance. This provides a very valuable starting point for the community to benchmark on this dataset.

### Weakness:

I have several questions that I hope the authors can answer:

- Is promptPG using rationale augmented few shot examples? What happens if it uses simple few-shot without rationales?

- What happens if we increase the number of shots in-context? Would using more than 2 shots reduce the variance that come from choosing different prompt examples?

- Clarify what it means to have or not have C (choice options) in Table 4.

- Can you train for a fixed set of in-context examples? What would be the performance gap? It would be interesting to see how much does the improvement come from dynamic prompt examples rather than an optimized set of static examples.

**Summary Of The Paper:**

This paper introduces TabMWP - a dataset containing open-domain grade-level problems that require mathematical reasoning on both textual and tabular data. It evaluates a number of models and methods on this dataset and finds that chain-of-thought prompting on GPT-3 is the strongest baseline. Inspired by the sensitivity of the chain-of-thought performance to the choice of prompt examples, the paper further introduces a method to learn to predict a dynamic set of prompt examples per test-question, where the model is trained through REINFORCE to output prompt examples that would maximize the expected accuracy on the test question.

**Summary Of The Review:**

This paper introduces a tabular math word dataset and benchmarks a number of methods on this task. The paper further introduces an RL-based method for dynamically choosing the prompt examples to be used for in-context learning. Both are novel and significant contributions to the best of my knowledge.

---

> ### Author Response · Authors · 2022-11-11
> **Response to Reviewer G2Gx (R3)**
>
> Dear reviewer, we really appreciate your great effort in reviewing our paper, and we thank you for your careful and helpful comments. We are very glad that you recognize that our dataset is well-designed, highly original, and extremely valuable to the community. We're encouraged to see that you think our model is novel, interesting, and makes significant contributions that do not exist in prior works.
>
> We address your questions and suggestions below. And we are looking forward to your further feedback to improve our paper in the discussion stage!
>
> > **Q1: Is PromptPG using rationale augmented few shot examples? What happens if PromptPG uses simple few-shot without rationales?**
>
> Yes, rationale augmented (chain-of-thought prompting, CoT) few-shot examples are used in our final PromptPG, which is represented as "Few-shot-CoT (2-shot)" in Table 3. Based on our experimental study, chain-of-thought prompting is very helpful for multi-hop reasoning tasks like TabMWP. Therefore, we built our final PromptPG on top of the few-shot-CoT GPT-3 model.
>
> When developing our policy gradient method, we also found that PromptPG gained greater advantages over random selection in the few-shot-CoT setting than in the few-shot setting without rationales. In the few-shot setting, the in-context examples only provide the task format information for the GPT-3 model. In contrast, in-context examples with multi-step solutions contain useful information about the reasoning process. With the help of our PromptPG, the GPT-3 model can highly benefit from the solutions if good examples are provided.
>
>
> > **Q2: What happens if we increase the number of in-context shots? Would using more than 2 shots reduce the variance that comes from choosing different prompt examples?**
>
> That is a good point, and thank you for your suggestion! We added the experiments for few-shot-CoT GPT-3 with random selection in terms of the different numbers of in-context shots. For each number of in-context shots, the experiment was conducted on 1,000 development examples and repeated three times. We report the average accuracy as we did in the submitted paper as follows:
>
> | Model              | Selection strategy | Shot number | Acc. (%) |
> | :----------------: | :---------: | :-----------------: | :------: |
> | Few-shot-CoT GPT-3 | Random selection    |  2 | 65.2 ± 4.01 |
> | Few-shot-CoT GPT-3 | Random selection    | 3 |  65.7 ± 1.16 |
> | Few-shot-CoT GPT-3 | Random selection    | 4 | 67.7 ± 0.78 |
> | Few-shot-CoT GPT-3 | Random selection    | 5 | 67.5 ± 0.98 |
> | Few-shot-CoT GPT-3 | **PromptPG (ours)** | 2 | **70.9 ± 1.27** |
>
> When increasing the number of in-context shots from the current 2 to 4, the few-shot-CoT GPT-3 model reduces the prediction variance from the random selection of in-context shots and achieves an accuracy improvement of 2.5%. When the number of in-context shots is increased to 5, the few-shot-CoT GPT-3 model with random selection does not gain further benefits. Our PromptPG displays impressive advantages over random selection in terms of data efficiency and prediction accuracy. With only two in-context shots, PromptPG achieves the highest accuracy of 70.9% and a comparable low deviation compared to random selection with more shots.
>
> > **Q3: What it means to have or not have C (choice options) in Table 4?**
>
> The input having C (choice options) means the choice options come after the question text, while the input not having C refers to no choice options being used in the input. Thanks for pointing it out! We have clarified it in the Table 4 caption in the revised paper now.
>
> > **Q4: What would be the performance gap if we train for a fixed set of in-context examples?**
>
> Thank you for your insightful suggestion! We achieved a new selection strategy, termed **manual selection**, which selects the two best examples from 20 as the fixed set of in-context examples. To be specific, we first randomly select 20 in-context examples from the training data. We evaluate each in-context example based on its one-shot-CoT performance on GPT-3 on 1,000 development examples. We choose the top two in-context examples with the highest evaluation accuracy and use them as the fixed set of in-context examples for the GPT-3 model. The new result is added in Table 5 of the revised version.
>
> We copied part of the results in Table 5 below:
>
> | Selection strategy | Acc. (%) |
> | :----------------: | :---------: |
> | Random selection | 65.2 ± 4.01 |
> | Manual selection | 66.9 ± 0.00 |
> | Nearest neighbor | 68.2 ± 0.29 |
> | **PromptPG (ours)** | **70.9 ± 1.27** |
>
> Manual selection can achieve the lowest prediction variance of 0 by selecting the two best examples from a fixed set of in-context examples. However, it only gains an improvement of 1.7% over random selection. Instead, our PromptPG learns to select instance-relevant in-context examples for each test example from a small set of 20 candidates and is able to achieve an improvement of 5.7% over random selection.

---

> ### Author Response · Authors · 2022-12-01
> **Thanks to Reviewer G2Gx**
>
> Dear reviewer,
>
> Please accept our sincere thanks again for all your suggestions on our work. We greatly appreciate your time and great efforts in improving our paper! We hope our responses have answered your questions. We are happy to answer any questions you may have later.
>
> Best regards,
>
> Authors of paper 828

---

> ### Author Response · Authors · 2022-12-05
> **Looking forward to a discussion before the deadline**
>
> Dear Reviewer,
>
> Thanks again for your great efforts in reviewing our paper!
>
> We have addressed all your questions in detail. As the deadline for the discussion is fast approaching, we are really looking forward to having a discussion with you on the OpenReview system. Would you mind checking our response and letting us know if you have further questions?
>
> With sincere regards,
>
> Authors of Paper 828

---

### Official Review · Reviewer_MYrE · 2022-10-27

**Confidence:** 3
**Clarity, Quality, Novelty And Reproducibility:** Novelty is debatable, but Clarity, Qu…
**Correctness:** 4
**Technical Novelty And Significance:** 3
**Empirical Novelty And Significance:** 2
**Recommendation:** 6

**Strength And Weaknesses:**

Strength
1. The paper is very well-written and easy to follow
2. the baseline evaluation of the proposed dataset is comprehensive

Weakness
1. The paper would benefit from a related work subsection that discusses if policy gradient has been applied in similar setup, or is this totally new?
2. There are a lot of math NLP datasets out there.  I understand this may be the first dataset that tests reasoning on both textual and tabular data, but it is unclear how this dataset would push forward NLP research.  Unlike text + image, there's already a quite natural way of combining table and text, so I'm not sure how this dataset may open up new research directions.  So this is a concern of the potential impact of this dataset.

**Summary Of The Paper:**

1. This paper proposes TabMWP, a dataset of 30k examples containing grade-level problems that require mathematical reasoning on both textual and tabular data.
2. The paper establishes a number of baselines for the dataset
3. The paper proposed a policy gradient method to select few-shot in-context examples

**Summary Of The Review:**

this is a pretty complete piece of work that has a new dataset and comprehensive experimental evaluation on the dataset.  whether this dataset can inspire important future research is unclear.

---

> ### Author Response · Authors · 2022-11-11
> **Response to Reviewer MYrE (R2)**
>
> Dear reviewer, thank you for the insightful comments, and we appreciate your time and effort.  We are excited that you evaluate our paper as a pretty complete piece of work that has a new dataset and comprehensive experimental evaluation with well-supported claims and statements. We appreciate that you recognize that our paper has good clarity, quality, and reproducibility and is very well-written. We are willing to address your concerns below.
>
> If you have any questions, please feel free to let us know. We are grateful to have the chance to work with you to improve the paper in the discussion phase!
>
> > **Q1: Suggestion: add a related work subsection that discusses if policy gradient has been applied in similar setups, or if is this totally new.**
>
> Thank you for your suggestion! We’ve added Subsection A.6 to review the applications of policy gradient in other tasks and point out the difference in our setup. We’d like to include the discussion here as well.
>
> Many policy gradient algorithms have been proposed in the past decade. They have been proven effective in areas like robotics and chatbots. In recent work that tries to align language models with human values, a policy gradient has been used to optimize language models with rewards learned from human feedback and preference. To the best of our knowledge, our PromptPG is the first work that proposes to select prompts dynamically for large pre-trained language models in the mathematical reasoning field.
>
>
> > **Q2: The potential impact of this dataset: how this dataset would push forward NLP research?**
>
> As supported by Reviewers R1-fX43, R3-G2Gx, R4-5Mut, and R5-MEHB, our proposed dataset is novel, high-quality, and well-designed, and is a valuable and significant contribution to the community on mathematical reasoning, tabular reasoning, natural language processing, and deep learning. There are multiple impacts of the dataset, as discussed below:
>
> First, TabMWP is a new, well-designed, high-quality benchmark for **mathematical reasoning**. The TabMWP dataset contains 38,431 math word problems with tabular contexts. TabMWP differs from previous MWP datasets in various aspects, including: (1) TabMWP is the first dataset to study math word problems over tabular context on open domains and is the largest in terms of data size; (2) TabMWP contains diversity in terms of question types (free text, MC), answer types (text span, integer number, decimal number), domains, and grade levels (grades 1-8); and (3) Each problem is annotated with natural language solutions to reveal multi-hop reasoning steps. It sets up a well-formed benchmark for machine learning methods: a system needs to understand the heterogeneous information of tubular and textual data, perform multi-hop reasoning by looking up table cells given textual clues, and generate multi-step operations to predict the final answer.
>
> Second, TabMWP poses new challenges for existing work on **table question answering**. Unlike Table QA datasets like FinQA, TAT-QA, and MultiHiertt, a lack of either mathematical reasoning or the tabular context renders the problems in TabMWP unanswerable. Furthermore, TabMWP features three formats for the tabular context, including an image, a semi-structured format, and a structured format. The diverse tabular formats enable TabMWP to be a general-purpose benchmark for different lines of approaches, including visual question answering methods, language models, and SQL-based methods.
>
> Third, TabMWP is a valuable benchmark for multi-modal and complex reasoning for the communities of **natural language processing** and **deep learning**. Each question in TabMWP is accompanied by a tabular context and requires multiple reasoning steps. Thorough experiments reveal that existing SOTA QA models like UnfiedQA and SOTA TableQA models like TAPEX fail to achieve satisfactory performance on TabMWP.
>
> Last, TabMWP is quite helpful for developing powerful **large language models** with diverse reasoning abilities. Each problem in TabMWP is annotated with high-quality natural language solutions with multiple steps. Therefore, TabMWP can be used to build and evaluate different prompting techniques such as chain-of-thought, least-to-most, and selection-inference prompting.

---

> ### Author Response · Authors · 2022-12-01
> **Thanks to Reviewer MYrE**
>
> Dear reviewer,
>
> Please accept our sincere thanks again for all your suggestions on our work. We greatly appreciate your time and great efforts in improving our paper! We hope our responses have answered your questions. We are happy to answer any questions you may have later.
>
> Best regards,
>
> Authors of paper 828

---

> ### Author Response · Authors · 2022-12-05
> **Looking forward to a discussion before the deadline**
>
> Dear Reviewer,
>
> Thanks again for your great efforts in reviewing our paper!
>
> We have addressed all your questions in detail. As the deadline for the discussion is fast approaching, we are really looking forward to having a discussion with you on the OpenReview system. Would you mind checking our response and letting us know if you have further questions?
>
> With sincere regards,
>
> Authors of Paper 828

---

### Official Review · Reviewer_fX43 · 2022-10-27

**Confidence:** 3
**Correctness:** 4
**Technical Novelty And Significance:** 2
**Empirical Novelty And Significance:** 3
**Recommendation:** 6

**Clarity, Quality, Novelty And Reproducibility:**

Clarity, quality and reproducibility are high.  As mentioned above, the novelty is less clear.

**Strength And Weaknesses:**

Strengths:

- The new dataset is a valuable resource for an interesting problem, and goes beyond similar previous datasets in terms of scale and types of problems as shown in Table 2.
- The PromptPG method demonstrates impressive performance, and the experimentation is very thorough in testing against SOTA approaches, the influence of hyperparameters, human performance, simpler selection strategies (random / nearest neighbor).

Weaknesses:

- The method is mainly a combination of previous approaches, and there is not substantial theoretical/conceptual novelty
- It is not clear why the method is framed specifically as a reinforcement leaning method - since the problem instances are all independent, it would seem more accurate to describe it as online learning, and describe the approach as optimizing a supervised model with latent variables.

**Summary Of The Paper:**

This paper introduces a dataset of Tabular Math Word Problems (TabMWP), consisting of Math word problems with associated tabular data.  Additionally, the authors introduce a policy gradient approach (PromptPG) for selecting in-context examples as prompts for a GPT-3 few-shot language model in generating answers to the TabMWP problems.  The dataset is open-source, and contains multiple types of answers (numeric, multiple-choice, text), and the policy gradient method is shown to outperform other approaches.

**Summary Of The Review:**

The paper represents a potentially valuable contribution to an interesting problem in natural language processing, showing how a 'high-level' algorithm can be developed by using existing components to achieve SOTA results in a surprisingly straightforward manner.  The paper will be of interest to those working in natural language processing and deep learning.

---

> ### Author Response · Authors · 2022-11-11
> **Response to Reviewer fX43 (R1)**
>
> Dear reviewer, thank you for the constructive comments, and we appreciate your time and effort. We are glad that you recognize that our dataset presents a valuable resource/contribution to an interesting problem and goes beyond similar previous datasets. It is helpful that you recognize our proposed method outperforms SOTA approaches over very thorough experimentation, and the paper will be of interest to the community. We address your questions below:
>
> > **Q1: The novelty and contribution of the method.**
>
> Prompting learning for large-pretrained language models (LLMs) like GPT-3 is a recently introduced problem and is still understudied after early work (Brown et al., 2020) shows LLMs in a few-shot setting have remarkable performance on downstream NLP tasks. The few-shot GPT-3 model highly depends on the selection of in-context examples, and its performance is unstable and can degrade to near chance in the worst-case scenario. To solve this problem, we propose a novel method, PromptG, the first work that applies reinforcement learning to learn to select in-context examples without any manually designed heuristics and shows impressive performance over various SOTA methods. To be more specific,
>
> (1) Recent work shows that few-shot GPT-3 is sensitive to the selection of in-context examples. In a pioneering attempt, our work investigates different selection strategies, including random selection, nearest neighbor research, and manually designed search strategies. We further propose PromptG, a policy gradient method that automatically learns to select good in-context examples from a small subset of training examples for the test example.
>
> (2) To the best of our knowledge, our PromptG is the first work that uses the RL method to learn good in-context examples for GPT-3 with a small amount of training data. Extensive experiments have shown that it largely reduces the prediction variance and achieves better performance over the SOTA baselines. It is supported by Reviewer-3 G2Gx, who claims that our method is novel, has significant contributions, and does not exist in prior works.
>
> (3) As the first attempt of this kind, we design a straightforward and extremely effective way to implement the reinforcement learning method for dynamical prompt learning. A comprehensive experimental comparison with existing methods and selection strategies verifies the impressive performance of our proposed method. It could be used as a plug-and-play module for a wide range of LLM-powered tasks.
>
> (4) We also investigate the effect of various parameters in our proposed method, including (a) the number of candidate examples, (b) the number of training examples, and (c) different selection strategies.
>
> (5) We further conduct an extensive case study, visualizing the predictions from three methods, as well as successful and unsuccessful cases of our PromptPG. This study is helpful in shedding light on the follow-up work to design more powerful language models for mathematical reasoning.
>
> > **Q2: The motivation of the method is framed as a reinforcement learning method, not as supervised learning or online learning.**
>
> Inspired by the success of large language models in a few-shot setting, we formulate the TabMWP task as an in-context learning problem, where the GPT-3 model completes the test example by generating the rationale and answer, demonstrated with few-shot examples. Given the test example, our goal is to design an effective algorithm capable of searching for good in-context examples to create the prompt.
>
> If we had reward values for possible prompts for each test example, we could use a **supervised learning** algorithm to design a prompt predictor, taking the problem as features and predicting the prompt with the best reward. However, there is no such annotation of reward values in TabMWP. Meanwhile, there are no explicit heuristics to determine how good a prompt is for a test example. Instead, we propose to use **reinforcement learning** to explore the best prompt for each test problem without any value annotations or manually designed heuristics.
>
> **Online learning** is usually used in situations where it is necessary to dynamically adapt one model to new patterns in the data as the data becomes available in sequential order. Although our approach shares some commonalities with online learning in terms of dynamically adapting to different problems, we are not studying the problem in a real-time setting as online learning does. Moreover, our policy network is trained on a subset of training data and is fixed in the evaluation stage. Strictly speaking, our studied problem falls into the category of **contextual bandits**, which is considered a basic single-step reinforcement learning problem.

---

> > ### Comment · Reviewer_fX43 · 2022-12-13
> > **Response**
> >
> > Thanks for your detailed responses.  The explanation of the method as falling into the category of contextual bandits as a single-step reinforcement learning problem was helpful - it may be worth mentioning this somewhere in the paper.

---

> > > ### Author Response · Authors · 2022-12-13
> > > **Thanks for your suggestion!**
> > >
> > > Dear reviewer,
> > >
> > > Thank you for taking the time to provide your valuable feedback on our paper. We really appreciate your suggestion to mention contextual bandits as a single-step reinforcement learning problem, and will definitely incorporate this information in the future revision (the paper revision function is closed for now).
> > >
> > > Thank you again for your thoughtful input!
> > >
> > > Sincerely,
> > >
> > > The authors

---

> ### Author Response · Authors · 2022-12-01
> **Thanks to Reviewer fX43**
>
> Dear reviewer,
>
> Please accept our sincere thanks again for all your suggestions on our work. We greatly appreciate your time and great efforts in improving our paper! We hope our responses have answered your questions. We are happy to answer any questions you may have later.
>
> Best regards,
>
> Authors of paper 828

---

> ### Author Response · Authors · 2022-12-05
> **Looking forward to a discussion before the deadline**
>
> Dear Reviewer,
>
> Thanks again for your great efforts in reviewing our paper!
>
> We have addressed all your questions in detail. As the deadline for the discussion is fast approaching, we are really looking forward to having a discussion with you on the OpenReview system. Would you mind checking our response and letting us know if you have further questions?
>
> With sincere regards,
>
> Authors of Paper 828

---

### Author Response · Authors · 2022-11-11
**General comments**

We would like to thank the reviewers for providing us with thoughtful comments and constructive feedback!

We are encouraged that our proposed **dataset** TabMWP is new/novel (R1-fX43, R2-MYrE, R3-G2Gx, R5-MEHB), high-quality (R4-5Mut, R5-MEHB), well-designed (R3-G2Gx), and contains diverse/multiple types (R1-fX43, R3-G2Gx). It is glad to see that TabMWP is recognized as a helpful/interesting/valuable/significant contribution to the community on tabular reasoning, mathematical reasoning, natural language processing, and deep learning (R1-fX43, R3-G2Gx, R4-5Mut, R5-MEHB).

We are pleased that our proposed **method** PromptPG is new (R5-MEHB), novel, and interesting (R3-G2Gx). ALL five reviewers recognize that the experimental evaluation is well-designed/comprehensive/thorough, and PromptPG demonstrates impressive performance and achieves SOTA results over a large set of baselines.

We appreciate that R3-G2Gx recognizes that, the TabMWP **dataset** is highly original (first of its kind) and extremely valuable to the community, the proposed **method** is novel and significant, the **paper** is beautifully written and easy to follow, and our work shows significant contributions that do not exist in prior works.

We have incorporated the feedback and highlighted the updates in blue in the revised paper. In this revised paper, we mainly have the following updates:
- We added an experiment for few-shot-CoT GPT-3 with random selection in terms of the different numbers of in-context shots in Table 9, as suggested by R3-G2Gx.
- We added a new selection strategy, manual selection, in Table 5, as suggested by R3-G2Gx and R4-5Mut.
- We clarified some statements and fixed some typos.

---

### Author Response · Authors · 2022-11-18
**Responses to reviewers’ comments and revised paper uploaded**

Dear Reviewers:

We would like to sincerely thank you for your insightful and constructive comments. These are extremely helpful to our paper. With respect to these comments, we have posted point-to-point and detailed responses along with our revised paper. Please feel free to let us know if you have any further questions.

Much appreciated!

Best regards,

*The Paper 828 Authors*

---

### Author Response · Authors · 2022-12-12
**Huge thanks to Reviewers**

Dear reviewers,

We truly appreciate the time and effort you have invested in reviewing our paper. We are delighted to hear that two reviewers believe we addressed their concerns.

As **today is the deadline for updating reviews and discussions**, we would still like the opportunity to address any additional questions or concerns you may have. If you have any further comments or suggestions, please do not hesitate to let us know. We will do our best to address any issues you raise.

Thank you again for your valuable input and we look forward to hearing from you.

Sincerely,

The Authors

---

### Decision · Program_Chairs · 2023-01-20

**Decision:**

Accept: poster

**Justification For Why Not Higher Score:**

Because the proposed Policy Gradient algorithm was not tested on non-tabular tasks.

**Justification For Why Not Lower Score:**

Because the introduced semi-structured reasoning task and proposed policy gradient algorithm are novel and interesting.

**Metareview: Summary, Strengths And Weaknesses:**

This paper proposes TabMWP which is the first tabular math word problem dataset. They further propose PromptPG which utilizes policy gradient to pick the best examples from the training data for a given test question. Both the dataset and the algorithm are interesting contributions. Therefore, I recommend acceptance. I suggest authors to improve the paper by evaluating the algorithm on at last one non-tabular task.

**Note From Pc:**

if the above contains the word "oral" or "spotlight" please see: "oral" presentation means -> notable-top-5% and "spotlight" means -> notable-top-25%. As stated in our emails, we are disassociating presentation type from AC recommendations

**Summary Of Ac-Reviewer Meeting:**

4 reviewers attended the meeting and pros/cons of the paper were discussed. These are some of the main points raised by reviewers:

G2Gx: This method can be very useful when you have many possible prompt examples.

MYrE: The proposed dataset is mediocre but the idea of choosing relevant examples with RL is great!!!

fX43: The authors' response to the question about reinforcement learning framing makes sense to me.

MEHB: The proposed method leads to significant performance gain which is important.

After discussions with reviewers, the consensus was to accept the paper.